



# Exploring the uncertainties in the aviation soot-cirrus effect

Mattia Righi[1], Johannes Hendricks[1], and Christof Gerhard Beer[1]

[1]Deutsches Zentrum für Luft- und Raumfahrt (DLR), Institut für Physik der Atmosphäre, Oberpfaffenhofen, Germany

**Correspondence:** Mattia Righi (mattia.righi@dlr.de)

**Abstract.** A global aerosol-climate model, including a two-moment cloud microphysical scheme and a parametrization for aerosol-induced ice formation in cirrus clouds, is applied in order to quantify the impact of aviation soot on natural cirrus clouds. Several sensitivity experiments are performed to assess the uncertainties in this effect related to (i) the assumptions on the ice nucleation abilities of aviation soot; (ii) the representation of vertical updrafts in the model; and (iii) the use of reanalysis data to relax the model dynamics (the so-called nudging technique). Based on the results of the model simulations, a radiative forcing from the aviation soot-cirrus effect in the range of $-35\ \mathrm{mW\,m^{-2}}$ to $13\ \mathrm{mW\,m^{-2}}$ is quantified, depending on the assumed critical saturation ratio for ice nucleation and active fraction of aviation soot, but with a confidence level below 95% in several cases. Simple idealized experiments with prescribed vertical velocities further show that the uncertainties on this aspect of the model dynamics are critical for the investigated effect and could potentially add a factor of about two of further uncertainty to the model estimates of the resulting radiative forcing. The use of the nudging technique to relax model dynamics is proved essential in order to identify a statistically significant signal from the model internal variability, while simulations performed in free-running mode and with prescribed sea-surface temperatures and sea-ice concentrations are shown to be unable to provide robust estimates of the investigated effect. A comparison with analogous model studies on the aviation-soot cirrus effect show a very large model diversity, with a conspicuous lack of consensus across the various estimates, which points to the need for more in-depth analyses on the roots of such discrepancies.

## 1 Introduction

The aviation sector contributes about 2.4% of the global anthropogenic $CO_2$ and is one of the fastest growing anthropogenic sectors, which makes it one of the key targets for mitigating the anthropogenic impact on climate (Lee et al., 2010; Grewe et al., 2017; Lee et al., 2021). In the last decades, civil aviation experienced a steady growth in activity, resulting in increasing $CO_2$ emissions at an average rate of $2\%\ \mathrm{yr^{-1}}$ between 1970 and 2012, further accelerating in recent years (2013–2018) to a $5\%\ \mathrm{yr^{-1}}$ rate (Lee et al., 2021). Most of the Shared Socioeconomic Pathways (SSPs; Riahi et al., 2017), developed in the context of the assessments by the Intergovernmental Panel for Climate Change (IPCC), project this growth in aviation emissions to continue until at least 2040 (Gidden et al., 2019), although the unexpected worldwide outbreak of the SARS-CoV-2 pandemic in the early months of 2020 might milden such increase (e.g. Forster et al., 2020).

In addition to the well-understood impact of $CO_2$ and the related mitigation measures (Fuglestvedt et al., 2008; Dahlmann et al., 2016), aircraft emit also a number of non-$CO_2$ components, whose climate impact is still uncertain (Grewe et al., 2017).





This concerns, for instance, the role of nitrogen oxides ($NO_x = NO + NO_2$), which control ozone formation and affect methane lifetime (Grewe et al., 2019), aerosol particles and their interactions with clouds (e.g., Gettelman and Chen, 2013; Righi et al., 2013; Penner et al., 2018), as well as the formation and growth of contrails and contrail cirrus (Burkhardt and Kärcher, 2011;
Chen and Gettelman, 2013; Bock and Burkhardt, 2016).

Among these various aviation effects, the impact of aviation soot on natural cirrus clouds has gained attention in recent years due to its potentially large climate impact, possibly exceeding the contribution of most of the aforementioned components, including $CO_2$ (Hendricks et al., 2005; Penner et al., 2018). Cirrus clouds cover about 30% of the globe and have an overall warming impact on the Earth radiative balance, as on average their longwave warming effect dominates over shortwave cooling
(Hartmann et al., 1992; Hong et al., 2016; Chen et al., 2000; Gasparini and Lohmann, 2016; Heymsfield et al., 2017). The aerosol-induced formation of ice crystals (IC) in cirrus clouds can occur either via homogeneous freezing of supercooled liquid solution aerosols or via heterogeneous freezing on the surface of a so-called ice nucleating particle (INP; Vali et al., 2015). During the formation of cirrus clouds, both mechanisms can occur and their competition for the available supersaturated water vapour has crucial effects on the microphysical structure of these clouds, since it controls the number concentration and the
size of IC, and hence their optical and radiative properties (Kärcher, 2017). Several aerosol types have been shown to act as INPs at cirrus temperatures, including mineral dust and soot (Möhler et al., 2005; Hoose and Möhler, 2012; Cziczo et al., 2013; Kanji et al., 2017). Since aircraft soot emissions are directly released at cirrus altitudes ($p \lesssim 400$ hPa), several studies focused on their impact on cirrus properties and the resulting climate effect (Hendricks et al., 2005, 2011; Liu et al., 2009; Penner et al., 2009; Gettelman and Chen, 2013; Zhou and Penner, 2014; Kärcher, 2017; Penner et al., 2018; Zhu and Penner, 2020; McGraw
et al., 2020).

Nevertheless, the above mentioned studies have indicated very large uncertainties on the magnitude, and even on the sign, of the impact of aviation soot on the radiative forcing (RF) exerted by natural cirrus. Part of these uncertainties derives from the assumptions on the still poorly understood ice nucleating properties of aviation soot. This concerns in particular the critical saturation ratio ($S_{crit}$) at which soot can initiate freezing and the active fraction ($f_{act}$) of the soot particle population that can
act as INP. Applying a global aerosol model coupled with a two-moment ice microphysical scheme (CAM3-IMPACT), Liu et al. (2009) estimated an aviation-soot effect of $-110$ mW m$^{-2}$ assuming a critical ice saturation ratio for soot nucleation $S_{crit} = 1.2 - 1.3$, but a positive RF of 260 mW m$^{-2}$ if soot is considered to be a worse INP with $S_{crit} = 1.4$. In both cases, a very high active fraction $f_{act} = 100\%$ was assumed for aviation soot. Using the same model, Penner et al. (2009) applied two different ice nucleation parametrizations (Kärcher and Lohmann, 2002; Liu and Penner, 2005) and quantified an aviation soot-
cirrus RF of $-161$ and $-124$ mW m$^{-2}$. In the first case, they assumed $S_{crit} = 1.3$ for soot, while in the latter the occurrence of heterogeneous freezing on soot particles was determined as a function of temperature. In both cases, the whole soot population was assumed to act as INP (i.e., $f_{act} = 100\%$). Using the ECHAM4 model and the Kärcher et al. (2006) parametrization for ice nucleation which is also applied in the present study, Hendricks et al. (2011) found no statistically significant climate effect when assuming that 10% of aviation soot acts as INP, with $S_{crit} = 1.2$. No statistical effects were also reported by Gettelman
and Chen (2013), who applied the CAM5 model with the assumption that soot has similar ice nucleating properties as mineral dust (i.e., $S_{crit} = 1.2 - 1.3$), but with lower (and presumably more realistic) active fractions. Zhou and Penner (2014) again used


the CAM5 model with the Liu and Penner (2005) ice nucleation parametrization, but provided an explicit calculation of the soot active fraction by considering its pre-processing in contrail cirrus. This resulted in 0.6% of aviation soot being an efficient INP, leading to a RF quantified in the range of $-350\ \mathrm{mW\,m^{-2}}$ to $90\ \mathrm{mW\,m^{-2}}$, depending on the assumptions on the background

sulfate and dust concentrations, both competing with soot for the formation of IC in cirrus clouds. Using the same model with the Kärcher et al. (2006) parametrization for ice nucleation and an improved scheme for subgrid-scale vertical updrafts, Penner et al. (2018) simulated an aircraft effect on cirrus clouds of $-200\ \mathrm{mW\,m^{-2}}$, considering $S_{\mathrm{crit}} = 1.35$ and again $f_{\mathrm{act}} = 0.6\%$ as a result of soot pre-processing in contrails. A subsequent study, using the CESM model featuring a hybrid ice nucleation scheme combining the best features of the Liu and Penner (2005) and Kärcher et al. (2006) parametrizations, updated this

estimate to $-140\ \mathrm{mW\,m^{-2}}$, assuming a slightly lower $S_{\mathrm{crit}} = 1.2$ (Zhu and Penner, 2020). In the most recent work on this subject, McGraw et al. (2020) used the CESM2 model with the cirrus nucleation scheme by Barahona and Nenes (2009) and found that aicraft soot can perturb cirrus clouds in the Norther Hemisphere, with effects both in the shortwave and in the longwave radiative forcing, but they were not able to extract a statistically significant signal from their model simulations.

     This brief literature review shows that the existing quantifications of the aviation soot-cirrus effect obtained with different

model approaches lead to very different results, ranging from statistically non-significant effects to potentially large radiative forcing values. The reason for this lack of consensus lies in the high complexity of the physical processes controlling this effect, which are hard to constrain by measurements and difficult to represent in global models (Kärcher, 2017). A few studies (Koehler et al., 2009; Mahrt et al., 2020) have shown that soot particles released by the combustion process in aircraft turbines could act as INPs at various ranges of atmospheric temperatures and ice supersaturations, but measurements are still limited.

To further complicate this picture, ice formation in the cirrus regime is also crucially driven by atmospheric dynamics and depends on the occurrence of vertical updrafts and on their strength (Kärcher and Podglajen, 2019; Kärcher et al., 2019). In spite of recent progress on the modelling side (Podglajen et al., 2016), the representation of vertical updrafts in global models is still subject to considerable approximations and does not capture small-scale variability in sufficient detail.

     In this work, we apply the EMAC model (ECHAM/MESSy Atmospheric Chemistry; Jöckel et al., 2010) with the aerosol

submodel MADE3 (Modal Aerosol Dynamics for Europe, adapted for global applications, third generation; Kaiser et al., 2014, 2019) in a recently-developed configuration (Righi et al., 2020) which includes a detailed parametrization for aerosol-induced ice formation in cirrus clouds (Kärcher et al., 2006; Kuebbeler et al., 2014). Rather than attempting to provide a single estimate of the aviation soot-cirrus effect, the goal of this study is to explore the uncertainties related to the microphysical and dynamic aspects of this effect, in order to provide a realistic, albeit broad, range of possible values for the resulting

climate impact. The microphysical analysis focuses on the ice nucleating properties of aviation soot, based on the results of the laboratory measurements reported in the literature. We show that the variation in both the critical saturation ratio $S_{\mathrm{crit}}$ for ice nucleation and the active fraction $f_{\mathrm{act}}$ of aviation soot can have a significant impact not only on the magnitude but also on the sign of the resulting RF. We attempt to relate this changes to the underlying physical processes as represented in the model. Furthermore, we analyse the role of the model's representation of vertical updraft by means of mechanistic studies in which

a very simple representation of such updraft is implemented in order to explore the sensitivity of the relevant microphysics to the model dynamics. This parametric approach has proven useful in a previous climate impact study (Righi et al., 2013), where





we analysed the uncertainties related to the assumptions on the size distribution of aerosol particles from different transport emission sources, including aviation. On a more general level, such methods were successfully used, for instance, to constrain the uncertainties on the microphysical properties of warm clouds (Lee et al., 2013) and on the aerosol indirect effect (Carslaw et al., 2013; Regayre et al., 2020).

This paper is organized as follows: in Sect. 2 the processes controlling the aviation-soot cirrus effect and its large uncertainties are discussed and the results of available laboratory studies are summarized. The model setup, which is largely based on Righi et al. (2020) but has been further improved here, and the performed numerical experiments are described in Sect. 3. The results are presented and discussed in Sect. 4 and we summarize the main outcomes of this study in Sect. 5.

## 2   Uncertainties in the soot-cirrus effect

As mentioned in the introduction, aerosol-induced ice formation in the cirrus regime ($T \lesssim -37°C$) can occur either via homogeneous freezing or by heterogeneous freezing on the surface of an INP. The latter process usually requires a lower critical supersaturation over ice and can therefore occur prior to the onset of homogeneous freezing, attenuating or even inhibiting the direct freezing of supercooled liquid solution aerosol. This has of course important consequences on the microphysical properties of cirrus, since it affects the number concentrations and size of IC and, in turn, the lifetime and the radiative properties of the clouds. Cirrus clouds generally exert large RFs, both in the shortwave and in the longwave spectrum, with the latter being slightly larger, which results in an overall warming effect (Hartmann et al., 1992; Hong et al., 2016; Chen et al., 2000; Gasparini and Lohmann, 2016). If a cirrus cloud is dominated by homogeneous freezing, adding more INPs typically results in a decrease of IC number concentration (ICNC) and a corresponding increase in their size (the so-called negative Twomey effect; Kärcher and Lohmann, 2003). Adding INPs to a cirrus cloud where heterogeneous freezing already dominates, on the other hand, could result in a further increase of ICNC and a decrease in crystal sizes. As shown by previous studies (e.g. Zhang et al., 1999), an increasing IC size reduces the longwave cloud forcing (i.e., less warming) but also the shortwave cloud forcing in absolute terms (i.e., less cooling), and their combination can be either a net warming or a net cooling depending on the cloud ice water content. Other effects, such as a more efficient sedimentation of less abundant, larger IC or an increased deposition of water vapour in the presence of more efficient INPs (dehydration; Jensen et al., 2001, 2013) add even more complexity to this picture and to the interpretation of the model results. For a reliable quantification of the effect of aviation soot on cirrus clouds and its climate impact, it is therefore essential not only to have a reliable estimate of the ice nucleating properties of aviation soot, but also of the dynamic processes that control the background state of natural cirrus clouds in the model. Cirrus formed in slow (fast) updraft are usually dominated by heterogeneous (homogeneous) nucleation, resulting in lower (higher) concentrations of IC and larger (smaller) IC sizes (Kärcher and Lohmann, 2002; Krämer et al., 2016; Krämer et al., 2020). Hence, the way aviation-soot INPs can impact on these clouds depends not only on their ice nucleation ability but also on the dynamical background conditions. In the present study we therefore focus on both the microphysical and the dynamic aspects, by analysing the uncertainties related to the assumptions on aviation soot INP characteristics and exploring the impact of different (simplified) representations of the dynamic forcing in the model. We stress again that the main goal of this study is



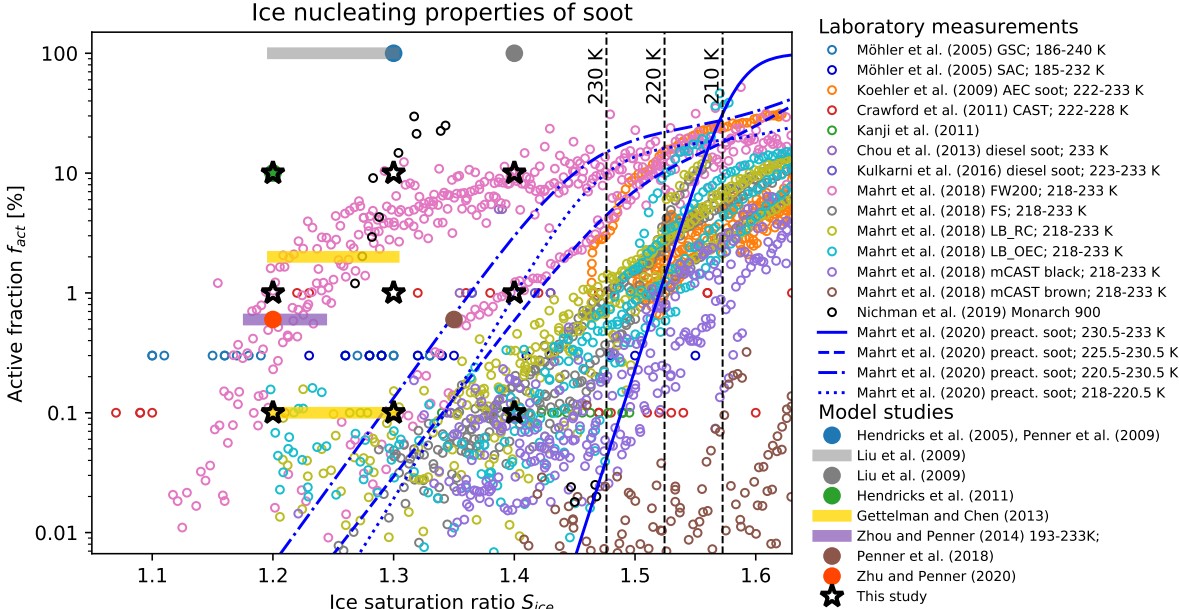

**Figure 1.** Summary of ice nucleating properties of soot measured in different laboratory studies (Möhler et al., 2005; Koehler et al., 2009; Crawford et al., 2011; Kanji et al., 2011; Chou et al., 2013; Kulkarni et al., 2016; Mahrt et al., 2018; Nichman et al., 2019; Mahrt et al., 2020) compared with the values assumed in model studies on the impact of aviation soot on cirrus (Hendricks et al., 2005; Penner et al., 2009; Liu et al., 2009; Hendricks et al., 2011; Gettelman and Chen, 2013; Zhou and Penner, 2014; Penner et al., 2018; Zhu and Penner, 2020). The parameters explored in this study are symbolized with stars. The vertical dashed lines show the homogeneous ice nucleation threshold at 210 K, 220 K and 230 K.

not to provide an updated estimate on the aviation soot-cirrus effect, but to explore its sensitivity to aviation soot microphysics and, to some extent, to the underlying model dynamics.

      Figure 1 summarizes the ice nucleating properties of different soot types retrieved from the literature, including cloud-processed soot. These properties are given in terms of ice saturation ratio $S_{\mathrm{ice}}$ and active fraction $f_{\mathrm{act}}$ and are compared with the assumptions made in the modelling studies on the aviation soot-cirrus effect reviewed in the introduction. Note that models

usually assume a critical ice saturation ratio $S_{\mathrm{crit}}$ at which ice nucleation takes place, while in the laboratory measurement a wide spectrum of $S_{\mathrm{ice}}$ values is explored. The measurements mostly report medium to low ice nucleation efficiency by different soot types ($S_{\mathrm{ice}} \gtrsim 1.3$) and activation fractions hardly above 10%. The modelling studies, on the other hand, tend to represent soot as a better INP, especially in terms of critical supersaturation ratio. Recent laboratories studies (Marcolli, 2017; Mahrt et al., 2020) support indeed higher ice nucleation efficiency for soot particles which experienced cloud processing,

for example in contrails, but this applies only to particles with diameters of about 400 nm, which implies that only a very low fraction of aviation soot can effectively be active as INPs in the upper troposphere. Further note that the experimental results show a clear temperature dependence on the soot ice nucleating properties, which is mostly ignored by the model



parametrizations. To explore this parameter space in sufficient detail, also reproducing the assumptions of previous model studies, we therefore perform nine sensitivity simulations in this study, varying $S_{\mathrm{crit}}$ from 1.2 to 1.4, and $f_{\mathrm{act}}$ from 0.1% to 10%.

These assumptions are marked with the black star symbols in Fig. 1 and result in nine combinations of these two parameters. Note that values $S_{\mathrm{crit}} > 1.4$ would mostly exceed the homogeneous freezing threshold at relevant cirrus temperatures and are therefore not worthy to be explored for the scope of the present study.

All previous model-based investigations on the aviation soot-cirrus effect considered approximated representations of the vertical updraft and its subscale variability: a common approach uses the square-root of the turbulent kinetic energy (TKE) as

a proxy for such variability (Lohmann and Kärcher, 2002). Later studies (Kuebbeler et al., 2014) also included the contribution of orographic gravity waves generated on the lee of mountain ranges (Joos et al., 2008), while recent models (Penner et al., 2018) used methodologies based on measurements (Podglajen et al., 2016) to consider the contribution of gravity waves to the vertical velocity. In this study, the aviation soot-cirrus effect is quantified with the EMAC-MADE3 model (see Sect. 3), which follows the TKE approach for the subscale vertical velocity, also considering the impact of orographic waves in relevant

regions. To further investigate how different cirrus regimes may react to the perturbation represented by aviation-soot INPs, we also consider an idealized representation of the vertical velocity. We prescribe constant values of the vertical velocity in the range from $2\,\mathrm{cm\,s^{-1}}$ to $50\,\mathrm{cm\,s^{-1}}$, to explore the full range of possible updraft regimes, including both slow and fast updrafts (Kärcher et al., 2006; Krämer et al., 2020). While this approach is of course idealized, it offers the possibility to separate the microphysical from the dynamic effect by artificially introducing a spatially uniform dynamic regime, thus allowing to

interpret the competition among the different INPs considered by the model purely in terms of their microphysical properties. Furthermore, it enables the investigation of INP effects under possible regimes, not covered by the TKE and orographic gravity wave approaches mentioned above.

## 3   Model description and simulations

The global aerosol model EMAC-MADE3 used in this study has been extensively evaluated in Kaiser et al. (2019), focusing

in particular on the representation of aerosol particles and their global distribution. In this work, we apply an advanced version of the model, including couplings of the aerosol submodel MADE3 with the radiation and cloud schemes. The model includes a two-moment cloud microphysical scheme based on Kuebbeler et al. (2014), which features a parametrization for aerosol-driven ice formation in cirrus clouds following Kärcher et al. (2006). This configuration has been tuned and evaluated with respect to the main cloud and radiation variables in Righi et al. (2020, hereafter R20). The work by R20 demonstrated the

overall ability of the model to reproduce the most important cirrus properties, like ice water content (IWC) and ICNC, when compared with in-situ measurements (Krämer et al., 2016; Voigt et al., 2017; Krämer et al., 2020), a necessary requirement in the context of the present work.

With respect to the EMAC-MADE3 version documented in R20, some technical changes and a few improvements specific to the application for this study have been introduced. In particular, the time integration of the cloud prognostic variables,

including cloud droplet and ice crystal number concentration, has been improved, to make it consistent with the approach used



by the other submodels which are part of the EMAC model system. This change required a retuning of the model configuration, which now uses slightly different values for the tuning parameters with respect to the R20 setup. More specifically, the autoconversion rate has been reduced from $\gamma_r = 8$ to $\gamma_r = 4$ and the minimum CDNC has been increased from $20\,\mathrm{cm}^{-3}$ to $50\,\mathrm{cm}^{-3}$. The overall model performance after this correction is similar to R20 and the main conclusions of that study remain valid: the

model provides a good representation of cloud cover, CDNC, precipitation and cloud radiative effects, while it is still affected by a relatively high LWP (at the high end of the observed range). The cirrus specific quantities, IWC and ICNC, show a slight improvement. Furthermore, the model now simulates an anthropogenic effective radiative forcing (ERF) of $-1.16\,\mathrm{W\,m}^{-2}$, in better agreement with the recent assessment by Bellouin et al. (2020), who reported a range $-1.6$ to $-0.6\,\mathrm{W\,m}^{-2}$ (68% confidence interval). The tuned model is characterized by a radiative balance of $3.4\,\mathrm{mW\,m}^{-2}$ when run in nudged mode, i.e.,

by relaxing the meteorology towards reanalysis data. The same configuration in free running mode has a radiative balance of $0.9\,\mathrm{mW\,m}^{-2}$ (see R20 for a detailed discussion on the impact of nudging on the model radiative balance).

    Another relevant improvement to the model configuration applied here is the introduction of an additional tracer BCtag to which the soot emissions from the aviation sector are assigned. The BCtag tracer is distributed into the same 6 modes as the standard BC tracer of MADE3, namely Aitken, accumulation and coarse mode, each with insoluble and mixed states. The

BC and BCtag tracers have the same physical properties and undergo exactly the same processes in the model, but allow for different ice nucleating properties between background and aviation soot in the cirrus parametrization. The ice nucleating properties of mineral dust and background soot are the same as in R20, namely $S_{\mathrm{crit}} = 1.1 - 1.2$ with a temperature-dependent active fraction for mineral dust in the deposition mode, $S_{\mathrm{crit}} = 1.3$ and $f_{\mathrm{act}} = 5\%$ for mineral dust in the immersion mode (Kuebbeler et al., 2014), and $S_{\mathrm{crit}} = 1.4$ and $f_{\mathrm{act}} = 0.25\%$ for background soot (Hendricks et al., 2011). The $S_{\mathrm{crit}}$ and $f_{\mathrm{act}}$

parameters for aviation (and in part also background) soot are explored in more detail in the dedicated sensitivity studies, as discussed in Sect. 2. To avoid confusion, we note here that the MADE3 BC and BCtag tracers actually refer to black carbon, i.e. an aerosol type composed only of carbon, but we are using the term soot in this paper for consistency with most of the literature on aviation effects, although these definitions are not fully consistent (see Petzold et al., 2013, for a more detailed discussion on this terminology).

To calculate the number concentration of INPs for the different types we use the same approach as R20, while for the newly introduced BCtag tracer we derive the number concentration from the tracer mass, by assuming aviation soot to follow the bimodal size distribution measured by Petzold et al. (1999) in the plume of a B737-300 aircraft. This distribution is characterized by median diameters of 25 and 150 nm, and geometric standard deviations of 1.55 and 1.65, for the Aitken and accumulation modes, respectively. The same size distribution parameters were used in Righi et al. (2013) to characterize

particle number emissions from aviation. Introducing the BCtag tracer has the advantage that a lower number of simulations needs to be performed to isolate the impact of aviation soot on cirrus clouds, since only two experiments are required for that, i.e. with and without the effect of the BCtag INPs in the cirrus parametrization. The difference between these two experiments hence provides an estimate of the resulting climate impact. The statistical significance of this estimate is also improved with respect to an approach where no tagging of aviation soot is included, since in that case four experiments would be required (with

and without aviation, with and without soot impact on cirrus) to isolate the effect. Another advantage of this tagging approach is





**Table 1.** Summary of the EMAC-MADE3 simulations performed in this study. $S_{\mathrm{crit}}$ and $f_{\mathrm{act}}$ refer to the ice nucleating properties of aviation soot, while $S_{\mathrm{crit}}^{\mathrm{bg}}$ and $f_{\mathrm{act}}^{\mathrm{bg}}$ indicate the ones of background soot (i.e., from other anthropogenic and biomass burning sources). Every simulation includes an extra spin-up year which is not considered for the analysis. The nudged simulations use meteorological renalysis data for the period 2001–2015.

| Name | $S_{\mathrm{crit}}$ | $f_{\mathrm{act}}$ [%] | $S_{\mathrm{crit}}^{\mathrm{bg}}$ | $f_{\mathrm{act}}^{\mathrm{bg}}$ [%] | Vertical velocity [cm s$^{-1}$] | Dynamics | Duration [years] |
|---|---|---|---|---|---|---|---|
| BASE | – | – | 1.4 | 0.25 | online | nudged | 15 |
| S12F01 | 1.2 | 0.1 | 1.4 | 0.25 | online | nudged | 15 |
| S12F1 | 1.2 | 1 | 1.4 | 0.25 | online | nudged | 15 |
| S12F10 | 1.2 | 10 | 1.4 | 0.25 | online | nudged | 15 |
| S13F01 | 1.3 | 0.1 | 1.4 | 0.25 | online | nudged | 15 |
| S13F1 | 1.3 | 1 | 1.4 | 0.25 | online | nudged | 15 |
| S13F10 | 1.3 | 10 | 1.4 | 0.25 | online | nudged | 15 |
| S14F01 | 1.4 | 0.1 | 1.4 | 0.25 | online | nudged | 15 |
| S14F1 | 1.4 | 1 | 1.4 | 0.25 | online | nudged | 15 |
| S14F10 | 1.4 | 10 | 1.4 | 0.25 | online | nudged | 15 |
| BASE-BG12 | – | – | 1.2 | 0.25 | online | nudged | 15 |
| S12F10-BG12 | 1.2 | 10 | 1.2 | 0.25 | online | nudged | 15 |
| BASE-V2 | – | – | 1.4 | 0.25 | 2 | nudged | 15 |
| BASE-V5 | – | – | 1.4 | 0.25 | 5 | nudged | 15 |
| BASE-V10 | – | – | 1.4 | 0.25 | 10 | nudged | 15 |
| BASE-V20 | – | – | 1.4 | 0.25 | 20 | nudged | 15 |
| BASE-V50 | – | – | 1.4 | 0.25 | 50 | nudged | 15 |
| S12F10-V2 | 1.2 | 10 | 1.4 | 0.25 | 2 | nudged | 15 |
| S12F10-V5 | 1.2 | 10 | 1.4 | 0.25 | 5 | nudged | 15 |
| S12F10-V10 | 1.2 | 10 | 1.4 | 0.25 | 10 | nudged | 15 |
| S12F10-V20 | 1.2 | 10 | 1.4 | 0.25 | 20 | nudged | 15 |
| S12F10-V50 | 1.2 | 10 | 1.4 | 0.25 | 50 | nudged | 15 |
| BASE-FREE | – | – | 1.4 | 0.25 | online | free | 30 |
| S12F10-FREE | 1.2 | 10 | 1.4 | 0.25 | online | free | 30 |
| S14F10-FREE | 1.4 | 10 | 1.4 | 0.25 | online | free | 30 |
| BASE-FREE-T | – | – | 1.4 | 0.25 | online | nudged (without T) | 15 |
| S12F10-FREE-T | 1.2 | 10 | 1.4 | 0.25 | online | nudged (without T) | 15 |
| S14F10-FREE-T | 1.4 | 10 | 1.4 | 0.25 | online | nudged (without T) | 15 |





that different ice nucleation abilities can be assumed for aviation soot and background soot, i.e. soot from background sources. The aviation soot-cirrus effect is estimated by calculating the difference between a given simulation and a baseline experiment (BASE), where aviation soot (i.e., the BCtag tracer) is not considered as INP in the cirrus parametrization. A paired sample t-test is applied to verify the null hypothesis that the annual mean values of a given quantity (e.g., RF) are identical in the two

simulations (with and without aviation-soot impact on cirrus). We express the response of the test in terms of confidence level, i.e., $100(1 - p)$, where $p$ is the $p$-value. Unless otherwise specified, we regard the results as statistically significant when the null hypothesis can be rejected at a confidence level larger than $95\%$ ($p < 0.05$).

Finally, to provide estimates of the aviation effects closer to the present day conditions, we updated the model emissions setup for anthropogenic and biomass burning (or open burning) sources of aerosol and aerosol precursor species to the recent

CMIP6 inventory for the year 2014 (van Marle et al., 2017; Hoesly et al., 2018; Feng et al., 2020). Aviation BC emissions in this inventory amount to about 10.5 Gg in 2014. Natural emissions (e.g., mineral dust, volcanoes, etc.) are considered as in R20, where mineral dust and sea spray emissions were calculated as a function of the local wind speed. For consistency, the prescribed mixing ratios of greenhouse gases used as input to the radiation scheme of EMAC (RAD; Dietmüller et al., 2016) are also updated to the 2014 values provided in Meinshausen et al. (2017) for the CMIP6 project, while for ozone we

use the geographically-resolved mixing ratios generated in support of CMIP6 by Hegglin et al. (in prep., see also Hegglin et al., 2016) for the same year. The simulations performed in this study are summarized in Table 1. They are all characterized by a T42L41 configuration, corresponding to a horizontal resolution of about $2.8° \times 2.8°$ and 41 non-equidistant vertical layers. The simulated time period covers the years from 2000 to 2015, with the year 2000 as spin-up and not included in the analysis. Unless otherwise specified, the model meteorology (temperature, winds and logarithm of the surface pressure)

is nudged towards the ERA-Interim reanalysis data (Dee et al., 2011) of the European Centre for Medium-Range Weather Forecast (ECMWF) for the same time period. Further sensitivity experiments are performed in free running mode to analyse the effect of nudging on the results and cover a period of 30 years, using prescribed climatological mean (2000–2009) sea-surface temperature (SST) and sea-ice concentration (SIC) from the Met Office Hadley Centre dataset (HadISST, Rayner et al., 2003).

The model's representation of vertical velocity follows the approach of Kuebbeler et al. (2014) as described in R20, but given its importance for the present study, we recall the main aspects here. The vertical velocity in the model is described as the sum of a large-scale and a sub-scale component. The latter is approximated according to Lohmann and Kärcher (2002) as $0.7\sqrt{\text{TKE}}$, while in the vicinity of mountain ranges the sub-scale contribution of orographic waves to the vertical velocity is considered instead, following the parametrization of Joos et al. (2008). The vertical velocities from these three components, as

simulated by EMAC-MADE3, are depicted in Fig. 2 for the BASE experiment, considering the 95[th] percentile of the vertical velocity distribution above 400 hPa. This allows to analyse the largest fluctuations in the vertical velocity distribution for each component. The large-scale component (Fig. 2a) is usually characterized by small updrafts of the order of a few centimeters per second and with little spatial variation. The additional sub-scale contribution from the TKE (Fig. 2b) emerges mostly in the Tropics, but it is generally below about $1 \, \text{cm} \, \text{s}^{-1}$, with the exception of the Himalayan and Andean mountain ranges, where

spikes of a few tens of centimeters per second can be expected. In these and other mountain regions, however, the model





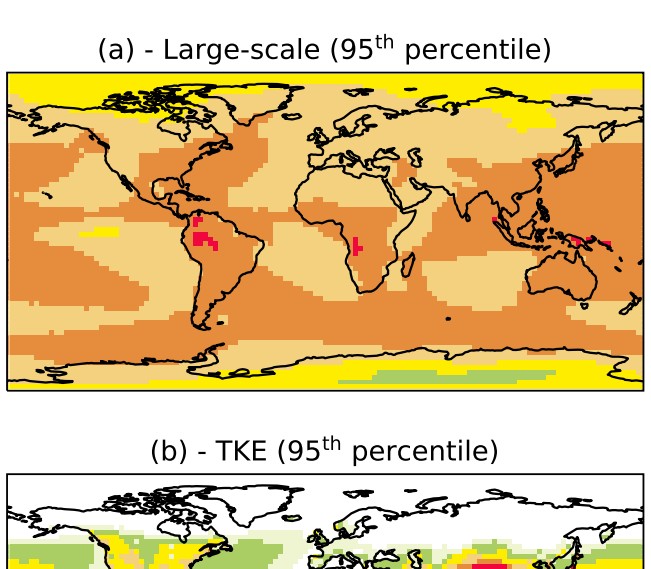

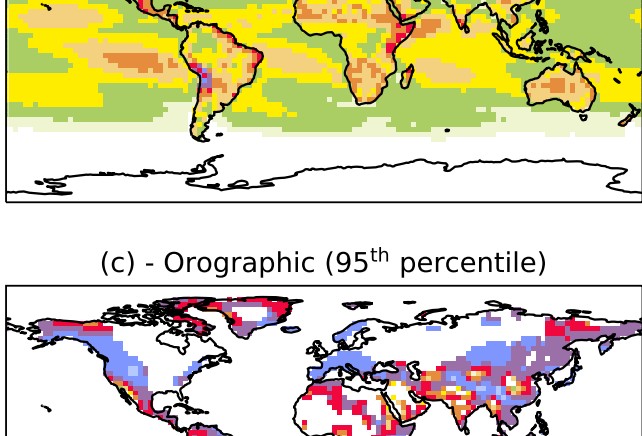

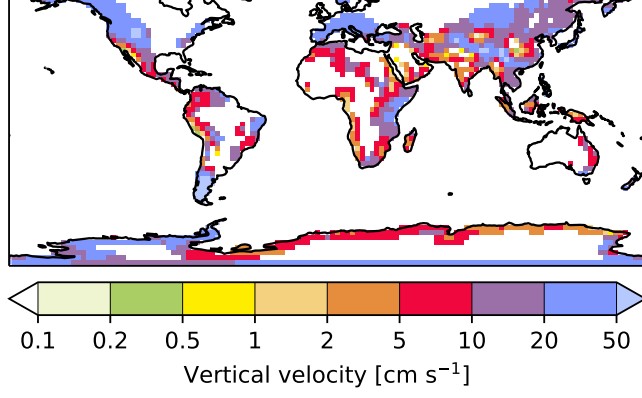

**Figure 2.** The vertical velocity components considered by the model: (a) large-scale; (b) sub-scale using the turbulent kinetic energy (TKE) as proxy; and (c) sub-scale contribution of the orographic waves in the vicinity of mountain ranges. Each panel shows the 95$^{th}$ percentile of the vertical velocity distribution with respect to the time (over the 2001–2015 simulation period) and vertical (above 400 hPa) coordinates.



accounts for the contribution of orographic waves, which can induce large vertical velocity fluctuations of about $50 \, \mathrm{cm \, s^{-1}}$, mostly impacting the western side of North America, Europe, and large parts of Asia around the Himalaya, i.e. near the most prominent mountain ranges of the world. Such strong updrafts can induce large supersaturations, thus efficiently driving homogeneous freezing and enhancing cirrus formation. The aviation effects investigated here could therefore be important in

these regions and the patterns depicted in Fig. 2 need to be taken into account for the interpretation of the results discussed in the next section.

## 4 Results and discussion

### 4.1 Geographical distribution of aviation soot and INPs

Before analysing the aviation-soot radiative effects, we present in Fig. 3 the geographical distribution of aviation soot emissions

from the CMIP6 dataset used here, together with the simulated mass and number concentration of aviation soot. Note that only a fraction of this number concentration is actually effective as INP in the model, depending on the $f_{\mathrm{act}}$ fraction assumed in the different sensitivity experiments discussed above. We stress again that the aviation emissions are assigned to a tagged soot tracer in EMAC-MADE3, thus making it possible to track aviation soot in the model without the need for an extra sensitivity experiment with aviation emissions switched off, as in the standard perturbation approach applied, for example, in Righi et al.

(2013). Not surprisingly, the aviation soot emissions are largest in the Northern mid-latitudes (Fig. 3a), with maxima in the Northern Hemisphere at typical flight altitudes ($200 - 250$ hPa) and close to the surface, due to the climb and descent phases, but also to the paths of short-range flights which are mostly common over the continents. The geographical pattern at flight altitude (Fig. 3a) shows the major routes connecting the most populated areas of the world, in particular the North Atlantic Flight Corridor between Europe and Eastern U.S., while the connections to Eastern Asia are less marked in this inventory.

The aviation soot mass concentration (Fig. 3a) follows a similar distribution, again with maxima at flight altitudes and close to the surface, with concentrations of the order of $0.2 - 0.5 \, \mathrm{ng \, m^{-3}}$. Not surprisingly, much lower mass concentrations are found in the Southern Hemisphere, again with a maximum at flight altitudes of $0.05 - 0.1 \, \mathrm{ng \, m^{-3}}$, at slightly higher level than in the Northern Hemisphere. This pattern is consistent with the results of a previous model study with the predecessor version of EMAC-MADE3 (Righi et al., 2013), where, however, lower concentrations of aviation soot were simulated. This

could be due to the use of a different emission inventory (CMIP5 instead of CMIP6) and for earlier conditions (year 2000 vs. 2014) in that study, but also to the introduction of the tagging method for aviation soot in the present work, which provides more accurate estimates. The difference in mass concentration between Northern and Southern Hemisphere is evident in the geographical map at $\sim 250$ hPa (Fig. 3b), where also a clear gradient from the Tropics towards the Polar regions is present. The number concentration of aviation soot particles (Fig. 3c,d) shows a similar pattern as the mass concentration, although with

clearer features near the emission sources, such as the North Atlantic Flight Corridor, and the regions above Central Europe and Eastern U.S. Some flight corridors can also be identified in the Southern Hemisphere. The reason for such sharply structured patterns is that particles in the Aitken mode, which dominate total particle number, are characterized by a shorter lifetime due to particle-particle interactions, which effectively reduce their number concentration away from sources, while their mass is of



**Figure 3.** Zonally averaged (a) and geographical distribution at 250 hPa (b) of aviation soot emissions from the CMIP6 dataset used for the model simulations in this work. The other panels show the mass concentration (c,d) and the number concentration (e,f) of aviation soot simulated by EMAC-MADE3 (BASE experiment, see Table 1). Panels (a), (c) and (e) show zonal means, whereas (b), (d) and (f) show the $\sim 250$ hPa level. All panels consider the multi-year average over the simulated period (2001–2015). Note that the number concentration of aviation soot is further multiplied by the active fraction $f_{act} = 0.1 - 10\%$ in the cirrus parametrization to obtain the number concentration of aviation soot INPs.

course conserved. For comparison, Fig. S1 in the Supplement shows the number concentration of the other INPs considered

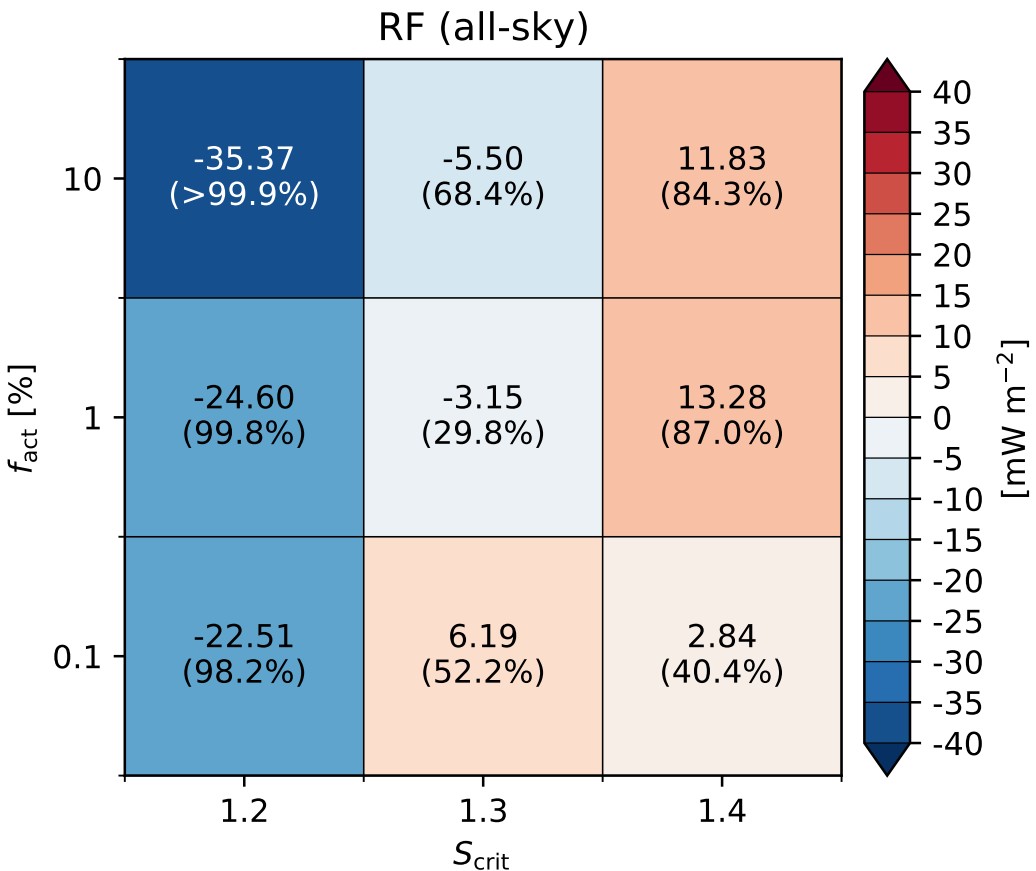

**Figure 4.** Multi-year average (2001–2015) top-of-the-atmosphere all-sky RF due to the effect of aviation soot on natural cirrus clouds, under different assumptions for the ice nucleation efficiency of aviation soot INPs ($S_{\mathrm{crit}}$ and $f_{\mathrm{act}}$). The values in brackets within each box indicate the confidence level.

by the model, namely mineral dust in the immersion and deposition mode, as well as background BC resulting from the other (non-aviation) emission sources.

## 4.2 The aviation soot-cirrus effect

The RFs from the aviation soot-cirrus effect under the nine different assumptions for the ice nucleating properties of aviation soot are presented in form of a matrix in Fig. 4. The EMAC-MADE3 simulations estimate this effect to be in the range of $-35\,\mathrm{mW\,m^{-2}}$ to $13\,\mathrm{mW\,m^{-2}}$, but lacking sufficient statistical significance for $S_{\mathrm{crit}} = 1.3$ and $S_{\mathrm{crit}} = 1.4$. The effect is negative (cooling) for low critical saturation ratios (higher nucleation efficiencies) and tends to increase towards a positive (warming) effect for medium to high critical saturation ratios (medium to low nucleation efficiencies). The statistical insignifi-





cance of the RF for some combinations of the parameters, however, makes it difficult to draw general conclusions from these overall numbers alone.

To facilitate the interpretation, we separate in Fig. 5a-d the different components of the RF, namely shortwave and longwave, all-sky and clear-sky. As expected, these graphs demonstrate how the aviation-soot cirrus effect actually results from the opposite changes in the shortwave (Fig. 5a) and longwave (Fig. 5b) all-sky RFs, which also have a much higher statistical significance than the total RF. The effect in the shortwave corresponds to a warming (i.e., a reduced cooling), meaning that aviation soot reduces the (background) cooling impact of natural cirrus clouds in the shortwave. This could be a manifestation

of the negative Twomey effect (Kärcher and Lohmann, 2003): the additional INPs from aviation compete with homogeneous freezing for the available supersaturation and lead to the formation of fewer and larger IC in these clouds, thereby decreasing their reflectivity and their shortwave cooling. This interpretation is consistent with the very small changes in the clear-sky part of the shortwave spectrum (Fig. 5c). Furthermore, the general decrease in the number concentration of homogeneously formed IC (Fig. 5e) seems to support this hypothesis, although the changes in overall ICNC are very small and not statistically

significant (Fig. 5f). The decrease in cloud frequency (Fig. 5h), could also contribute to these changes in the shortwave RF, especially at higher $S_{\mathrm{crit}}$, where the competition bewteen homogeneous and hetereogenous freezing which drives the negative Twomey effect can be expected to be less important. The warming effect in the shortwave spectrum is counteracted by a cooling (i.e., a reduced warming) in the longwave (Fig. 5b), due to the fact that larger IC usually sediment more efficiently and reduce the lifetime of the cirrus clouds, resulting therefore in a decrease of their (background) warming effect and hence in a cooling.

This is supported by the overall decrease in cloud frequency shown in Fig. 5h. The cooling (reduced warming) effect in the longwave is further enhanced when the nucleation efficiency of aviation soot is high ($S_{\mathrm{crit}} = 1.2$): in this case also the clear-sky RF (Fig. 5d) significantly contributes to the cooling. A possible aviation-induced (or aviation-enhanced) dehydration of the affected air masses is suggested here, resulting from the increased deposition of supersaturated water vapor on the very efficient INPs from aviation, which are then rapidly removed via sedimentation (Jensen et al., 2001, 2013). This reasoning is

supported by the marked decrease in total water (sum of water vapour and ice water, Fig. 5g) and in cloud frequency (Fig. 5h): here small ($\lesssim 1\%$) but statistically significant decreases in the simulations with $S_{\mathrm{crit}} = 1.2$ are evident. For lower ice nucleation efficiencies ($S_{\mathrm{crit}} \geq 1.3$), the changes are about one order of magnitude smaller and mostly not statistically significant. We can therefore conclude that the aviation-soot INPs can effectively enhance the dehydration of the upper troposphere and induce a statistically significant cooling effect when high nucleation efficiencies are assumed. For lower efficiencies (higher critical

saturation ratios), the warming (reduced cooling) in the shortwave, possibly due to enhanced cloud lifetime (Fig. 5h), appears to be more important and leads to a moderate overall RF effect.

The above analysis is based on globally averaged values, but large regional variations can be expected. This is because of the uneven geographical distribution of aviation-soot and other INPs (Figs. 1 and S1) and of cirrus clouds, but also because of the large spatial variations in the vertical velocity simulated by the model as shown in Fig. 2. As discussed above, the latter

effect is particularly important, as it controls the prevalence of homogeneous over heterogenous ice formation regimes and therefore the properties of natural cirrus clouds. The regional variations of the aviation-soot RF are depicted in Fig. 6, where the RF effect is separated in three different latitude bands, namely Southern Extratropics (SH-Ext, 30°S–90°S), Tropics (Trop,




**Figure 5.** As in Fig. 4, but for (a) all-sky shortwave; (b) all-sky longwave; (c) clear-sky shortwave; and (d) clear-sky longwave top-of-the atmosphere RFs. Panels (e-h) depict the aviation-soot-induced relative changes in ICNC from homogeneous freezing, ICNC, total water (as the sum of water vapour and ice water), and cloud frequency, respectively, all spatially averaged above 400 hPa and over cloudy and cloud-free model grid-boxes.

30°S–30°N) and Northern Extratropics (NH-Ext, 30°N–90°N), and compared to the global values as presented in Fig. 4. For high nucleation efficiencies ($S_{\mathrm{crit}} = 1.2$, Fig. 6a,d,g), the net cooling effect is common to all regions and particularly strong

in the Northern Hemisphere, where the aviation traffic is most intense and aviation soot shows the largest values of mass and number concentrations (Fig. 1). This cooling effect in the Northern Extratropics decreases gradually with decreasing ice nucleation efficiency $S_{\mathrm{crit}}$. For low efficiencies ($S_{\mathrm{crit}} = 1.4$, Fig. 6c,f,i) the warming effect from the Southern Extratropics and





Tropics dominates and results in a global warming effect. A reason for this pattern could be the smaller mean vertical velocities and the relatively clean background compared to the Northern Hemisphere. The additional aviation-emitted soot in this region could lead to enhanced heterogeneous nucleation and, due to the smaller cooling rates, to less homogeneous freezing, thus strengthening the negative Twomey effect.

As above, decomposing the RF effect in its different parts helps to disentangle the physical reasons for these aviation-induced effects on the RF. This is shown in Figs. S2–S5 in the Supplement. The warming effect in the shortwave (Fig. S2) is characterized by a very noisy pattern, with a prevalence of strong local warming effects especially in the Northern Extratropics, although the generally low confidence level ($< 90\%$) of the results hampers the identification of coherent patterns. The low values of the clear-sky RF (Fig. S4) confirm the dominance of cloud effects in the shortwave. The longwave effect (Fig. S3) shows a distinct and strong cooling over the continental regions of the Northern Extratropics, especially at high $f_{\mathrm{act}}$, with a slight dependency on $S_{\mathrm{crit}}$. This pattern very closely matches the ones of the orographic vertical velocity (Fig. 2c). In these regions, homogeneous ice formation is therefore expected to dominate the total ICNC, while aviation-soot can effectively compete against this process for the available water vapour. Its impact appears to be very effective regardless of the critical supersaturation, as long as the latter remains below the homogenous freezing threshold and a sufficient fraction (i.e. 10%) of aviation soot particles can be active as INPs. The consequence of this is a marked decrease in the cloud frequency (Fig. S6), which then results in the reduced longwave warming. Of course, the shortwave radiation component (Fig. S2) is also affected, and indeed shows a warming over the continents, but the signal is very noisy and not as evident as in the longwave. A possible explanation for this could be that other shortwave forcers like low-level clouds are contributing to the model variability, hence enhancing the noise, while in the longwave the cloud radiative effects are mostly due to cirrus clouds only. At high nucleation efficiencies ($S_{\mathrm{crit}} = 1.2$), a significant cooling is evident also in the clear-sky longwave RF (Fig S5), with a pretty uniform distribution in the extratropics, possibly due to a dehydration effect leading to a reduction in water vapour concentration and a resulting decrease in water-vapour-induced warming. The clear-sky longwave effect rapidly disappears at higher $S_{\mathrm{crit}}$, and in the Northern-Extratropics it turns to a warming, thus contributing to the decrease of the RF with $S_{\mathrm{crit}}$ in this region.

The results of our simulations show therefore that the largest (in absolute terms) and most statistically significant effect is simulated for large efficiencies of aviation soot, i.e. $S_{\mathrm{crit}} = 1.2$ and $f_{\mathrm{act}} = 10\%$, resulting in a cooling effect of $-35\,\mathrm{mW\,m^{-2}}$. Good nucleation abilities for aircraft soot ($S_{\mathrm{crit}} \simeq 1.2$) could be considered realistic for soot particles undergoing cloud pre-processing, e.g. in contrails. Laboratory measurements (Mahrt et al., 2020) show, however, that only large soot particles with a diameter above $\sim 400\,\mathrm{nm}$ may gain improved ice nucleation abilities resulting from such a pre-processing effect, but particles of this size are rare in the upper troposphere. Considering the measured size distribution by Petzold et al. (1999) as in Sect. 3, one can estimate the fraction of particles with diameter larger than 400 nm to be of the order of 0.001%, i.e. two orders of magnitude below the lowest $f_{\mathrm{act}}$ considered in this study (0.1%). The parameters measured by Petzold et al. (1999) refer to a young plume, so aging processes and soot aggregation within contrails might contribute to increase the fraction of larger particles in the population, but it is unlikely that a significant fraction of aviation soot will end up in the size range where pre-processing is effective, as also confirmed by aircraft measurements of ice residuals in cirrus and contrail cirrus (Voigt et al., 2017).



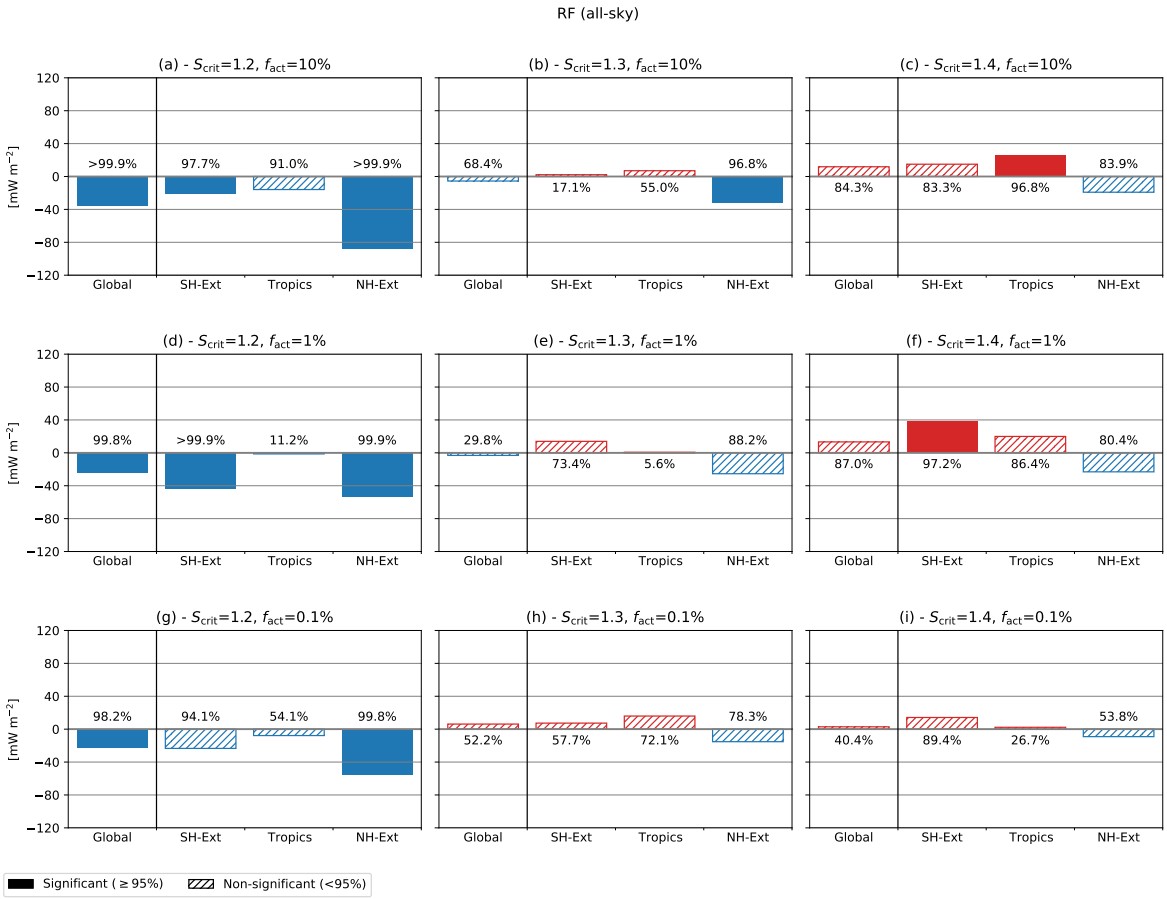

**Figure 6.** As in Fig. 4, but averaged over different regions. Each of the panels (a-i) corresponds to a given assumption for ice nucleation efficiency of aviation soot particles ($S_{crit}$ and $f_{act}$) as shown on the top. The values above or below each bar indicate the confidence level. Significant and non-significant results are further marked using filled and hatched bars, respectively. To facilitate the comparison, the bar in the left box in each panel shows the global RF values as in Fig. 4.

Our results generally point towards a relatively small aviation-soot cirrus effect, of the order of ten $mW\,m^{-2}$ (in absolute terms), with statistically non-significant figures in several cases. This is in contrast with the estimates by Zhou and Penner (2014), Penner et al. (2018) and Zhu and Penner (2020), who reported larger effects, of the order of hundred $mW\,m^{-2}$ (in absolute terms), also testing various assumptions on the ice nucleation abilities of aviation soot and experimenting with different parametrizations for ice nucleation. Due to the high complexity of the involved models and the coupling between their different components (aerosol, clouds, radiation and dynamics), it is challenging to track down the reasons for this disagreement, which could be due not only to different models' schemes and parametrizations, but also to the diverse configurations and tuning approaches. For example, Gettelman and Chen (2013) used a similar model (CAM5) as the aforementioned studies, but found no statistically significant effect of aviation soot on natural cirrus clouds, thus being more consistent with the results presented





here. McGraw et al. (2020) used a model version from the same family (CESM2, which is based on CAM6 for the atmospheric component), also concluding that the impact of aviation soot is not statistically significant. This was also the case in Hendricks et al. (2011), who used an ECHAM-based GCM as the present one and the same cirrus clouds parametrization, albeit with
different aerosol and cloud microphysical schemes. In conclusion, a consensus among modelling groups on the aviation-soot cirrus effect is still lacking and future research should consider working towards a concept for a model intercomparison study with common assumptions and detailed analyses of the differences among model configurations and tuning approaches.

    We finally recall that the sensitivity experiments conducted in this section are focusing on the ice nucleation abilities of aviation soot, while the properties of background soot (i.e., soot originating from other anthropogenic and biomass burning
sources) are not varied and are assumed to be $S_{\mathrm{crit}} = 1.4$ and $f_{\mathrm{act}} = 0.25\%$ as in Hendricks et al. (2011, see also Sect. 3). As a further sensitivity study, we perform two simulations assuming $S_{\mathrm{crit}} = 1.2$ for background soot (simulations BASE-BG12 and S12F10-BG12, see Table 1). This experiment pair results in an aviation soot-cirrus effect of $-25.7\,\mathrm{mW\,m^{-2}}$ (99.2% confidence level), which is lower (in absolute term) than the corresponding case calculated above with $S_{\mathrm{crit}} = 1.4$ for background soot (and $S_{\mathrm{crit}} = 1.2$ for aviation soot in both cases). This was to be expected, since increasing the ice nucleation abilities of background
soot enhances the competition with aviation soot and the other INPs for available water vapour, thus reducing the impact of aviation soot on natural cirrus clouds. The properties of background INPs could therefore be a further source of uncertainties for the aviation-soot cirrus effect and will be the subject of a companion study.

### 4.3   Dependency of the aviation soot-cirrus effect on the model representation of the vertical velocity

Besides the ice nucleating properties of soot, another major source of uncertainties in model studies attempting to quantify the
climate impact of aviation on cirrus clouds is the representation of vertical velocities. In the cirrus parametrization adopted here (Kärcher et al., 2006), the vertical velocity is a key parameter, as it controls the critical supersaturation, the competition between homogeneous and heterogeneous freezing, as well as the nucleation rate in cirrus clouds. Ice formation in cirrus clouds is strongly influenced by small scale updrafts, of the order of $1\text{--}10\,\mathrm{cm\,s^{-1}}$ (Barahona et al., 2017), but due to their coarse spatial resolution, global models are not able to represent such phenomena in sufficient detail, and rough approximations are usually
introduced to account for them. As explained in Sect. 3, in the EMAC-MADE3 configuration adopted here, the subscale vertical velocity is accounted for by adding an extra-term proportional to the square root of the turbulent kinetic energy to the large-scale, grid-box mean, vertical velocity. In the vicinity of mountain ranges, this term is replaced by the contribution of orographic waves to small-scale fluctuations in the vertical velocity, based on the parametrization by Joos et al. (2008), which could lead to stronger updraft of the order of several tens of centimeters per second.

To further explore the limitations behind this approach and their possible impact on the estimated aviation-cirrus effects presented in Sect. 4.2, we perform an additional set of sensitivity experiments, prescribing a geographically uniform constant vertical velocity in the range from 2 to $50\,\mathrm{cm\,s^{-1}}$. Such an assumption is of course not realistic, but the goal here is not to provide a refined estimate on the aviation soot-cirrus effect, but rather to understand the role of dynamic forcing on the results and hence to estimate the uncertainties associated with the RF effects quantified in the previous section. A prescribed constant
uniform field allows, for example, to explore regions of the world which could be important for the cirrus effect, but where





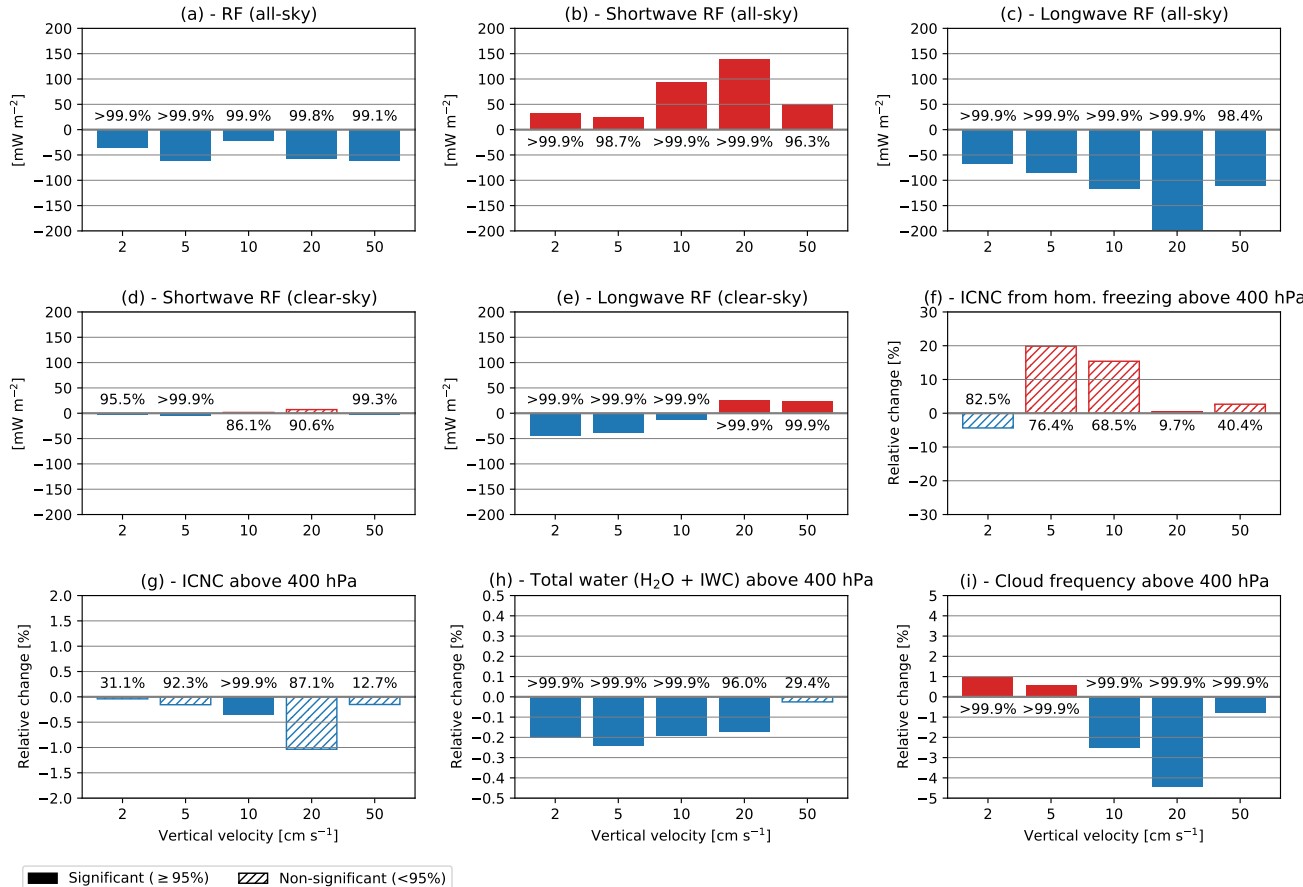

**Figure 7.** Multi-year average (2005–2015) TOA (a) all-sky; (b) all-sky shortwave; (c) all-sky longwave; (d) clear-sky shortwave; and (e) and clear-sky shortwave RF due to the effect of aviation soot on natural cirrus clouds, for different values of the vertical velocity. Panels (f-i) depict the aviation-soot-induced relative changes in ICNC from homogeneous freezing, ICNC, total water (as the sum of water vapour and ice water), and cloud frequency, respectively, all spatially averaged above 400 hPa and over cloudy and cloud-free model grid-boxes. The values above or below each bar indicate the confidence level. Significant and non-significant results are further marked using filled and hatched bars, respectively.

the model does not simulate a sufficiently strong updraft for ice formation to occur. Due to the large amount of computational resources required, we restrict this sensitivity analysis to a single pair of assumption for the ice nucleation abilities of aviation soot. To facilitate the analysis, we select the case $S_{crit} = 1.2$ and $f_{act} = 10\%$, which is more likely to return statistically significant results, as demonstrated in Fig. 4.

The simulated aviation-soot cirrus effect as a function of different prescribed vertical velocities is depicted in Fig. 7 in terms of RF (and its different components) and other relevant cloud properties, whereas Fig. S7 depicts the geographical distributions of the same quantities. The all-sky RF (Fig. 7a) remains negative for all explored values of the prescribed vertical velocity.





The aviation-soot cirrus RF for a prescribed vertical velocity of $2 \mathrm{~cm \, s^{-1}}$ (simulation S12F10-V2) is $-35.9 \mathrm{~mW \, m^{-2}}$, very close to the value of $-35.4 \mathrm{~mW \, m^{-2}}$ obtained with the parameterized vertical velocity (simulation S12F10). This suggests that the aviation effect discussed in this study is mostly controlled by small updrafts of a few centimeters per second. The RF is significantly larger, around $-60 \mathrm{~mW \, m^{-2}}$, for larger prescribed vertical velocities, except at $10 \mathrm{~cm \, s^{-1}}$, where a smaller value is simulated. This behaviour results from the combination of the shortwave (Fig. 7b) and longwave (Fig. 7c) all-sky RF, both increasing in absolute terms to a maximum value at $20 \mathrm{~cm \, s^{-1}}$ and significantly dropping above that value. This trend characterizes also the longwave clear-sky RF (Fig. 7e), which grows towards larger values with increasing vertical velocity and switches from cooling to warming at $20 \mathrm{~cm \, s^{-1}}$, while the shortwave clear-sky RF (Fig. 7d) is generally small across the explored range of vertical velocities. The geographical maps (Fig. S7) reveal that the pattern of the RF closely follows the distribution of aviation soot particles given in Fig. 3, with the largest values above the North Atlantic, Europe and Western U.S., especially for larger vertical velocities. These maps further reveal that the change of sign in the longwave clear-sky RF at $v > 10 \mathrm{~cm \, s^{-1}}$ is driven by an increase in total water in the Northern Hemisphere. Most of the aviation-soot induced changes in the other cloud parameters show a similar behaviour, with the maximum effects (in absolute terms) simulated at $v = 20 \mathrm{~cm \, s^{-1}}$. At larger vertical velocities, homogeneous freezing become very effective and rapidly consumes the available supersaturated water vapour, so that heterogeneously formed ice crystals have less time too grow and the sedimentation process becomes less effective. As a consequence, in this somewhat extreme regime the impact of aviation soot on ICNC, total water and cloud frequency (Fig. 7f-h) loses importance and results in smaller aviation-induced changes in the radiative fluxes.

The sensitivity simulations analysed in this subsection confirm that the model dynamics and the representation of the vertical velocity play an essential role in the microphysics of cirrus clouds. Even relatively small updrafts of a few centimeters per second can induce large changes in the properties of cirrus clouds perturbed by aviation emissions and that the resulting estimates on the climate impact are equally sensitive to the sub-scale fluctuations in the vertical velocity as to the ice nucleating properties of aviation soot particles. Our sensitivity experiments further show that the vertical velocity mostly controls the magnitude of the aviation-soot cirrus effect, while the ice nucleating properties of aviation soot also act on the sign. Both components still represent the largest source of uncertainties for all modelling studies attempting to estimate the climate impact of aviation soot on natural cirrus clouds.

### 4.4 Impact of nudging

All the results discussed so far refer to model simulations performed in nudged mode, i.e. relaxing meterological variables (temperature, divergence, vorticity and surface pressure) towards reanalysis data. This approach has been chosen in order to maximize the chances to obtain statistically significant results for the small climate effects and radiative forcings investigated here and it is a common practice in this kind of studies. This technique ensures that the simulations to be compared (in this case with and without aviation effect on cirrus) are characterized by a similar synoptic situation. Due to the chaotic nature of the climate system, this is not the case when running the model in free (climate) mode, as each experiment will develop its own meteorology, which hinders the isolation of the effect of a small perturbation such as the one represented by the impact of aviation soot on cirrus clouds. However, nudging is known to have an impact on simulated temperature profiles (Schultz



et al., 2018), in turn affecting the heating rates and all kinds of radiative adjustments in the atmosphere, which implies a potential influence on the effective radiative forcing of the climate perturbation under consideration (e.g., Forster et al., 2016; Johnson et al., 2019). In an attempt to characterize the impact of nudging on our results, we repeat the BASE, S12F10 and

450 S14F10 simulations in free running mode (BASE-FREE, S12F10-FREE and S14F10-FREE, respectively) and in nudged mode but without relaxing temperature (BASE-FREE-T, 12F10-FREE-T and S14F10-FREE-T, respectively, see Table 1). In these simulations, short term feedbacks on temperature (and in the FREE case on other dynamical quantities) can freely evolve. To establish climate conditions comparable to the years modelled in the nudged simulations, SST and SIC are prescribed in both sets: the FREE-T experiments use SST and SIC from the ERA-Interim reanalysis (i.e, the same dataset used for nudging winds

and surface pressure), while in the FREE ones climatological means of the 2000–2009 period from the Met Office Hadley Centre dataset (HadISST, Rayner et al., 2003) are used. To increase the chance of obtaining a statistically significant result, the FREE simulations are performed on a longer time period (30 years), while the FREE-T ones cover the same 15-year period as the nudged case.

The resulting all-sky RFs from the aviation soot-cirrus effect are compared in Fig. 8 for the three configurations (nudged,

nudged without temperature and free) and the two values of the $S_{\mathrm{crit}}$ parameter. As expected, the free running simulations are characterized by a much larger statistical noise, which prevents to draw any robust and statistically significant conclusion on the investigated effect and supports our choice for a nudged configuration. The results of the nudged experiments are, however, consistent with the free running ones, since they lie within the uncertainty ranges. The two nudging methods (with and without temperature) are highly consistent for $S_{\mathrm{crit}} = 1.2$, which further support the robustness of the results dicussed in

this work, although a feedback of temperature nudging on the dynamics cannot be excluded. For $S_{\mathrm{crit}} = 1.4$, the two nudging methods show RFs with opposite sign, although both are statistically non-significant at the 95% confidence level, therefore no conclusions can be drawn in this case. Nevertheless, this sensitivity study confirms the main conclusions of this work, that large aviation soot-cirrus effects can be simulated only under the assumptions of good ice nucleation ability of aviation soot particles ($S_{\mathrm{crit}} = 1.2$).

## 5 Conclusions and outlook

In this work, we applied the EMAC-MADE3 global aerosol climate model coupled with a cloud microphysical scheme to quantify the impact of aviation soot on natural cirrus clouds, and to estimate the uncertainties in this effect due to the assumptions on the ice nucleating properties of aviation and background soot particles and on the representation of vertical updrafts in the model. According to our model simulations, the aviation-soot cirrus effect is in the range of $-35\,\mathrm{mW\,m^{-2}}$ to $13\,\mathrm{mW\,m^{-2}}$,

with a confidence level $> 99.9\%$ and $87\%$, respectively, depending on the assumptions on the critical ice saturation ratio and on the fraction of active INPs for aviation soot. Further sensitivity experiments performed with a prescribed, geographically uniform, vertical velocity suggest that the uncertainties in the model representation of vertical updrafts and the resulting cooling rates can add a factor of $\lesssim 2$ to the estimated effect, although this has been tested only for a specific case corresponding to high nucleation efficiency of aviation soot ($S_{\mathrm{crit}} = 1.2$ and $f_{\mathrm{act}} = 10\%$).

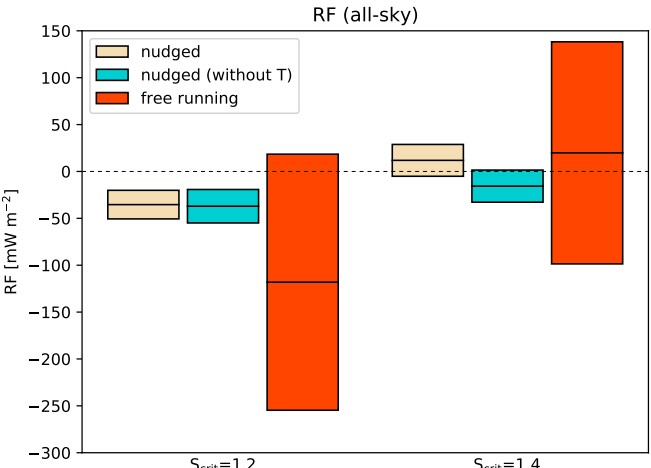

**Figure 8.** Multi-year average top-of-the-atmosphere all-sky RF due to the effect of aviation soot on natural cirrus clouds, under different model configurations: nudged, nudged without relating temperature, and free running. The latter configuration covers a period of 30 years, while the nudged ones cover 15 years. The horizontal line at the center of each box is the average value and each box spans the 95% confidence interval. Values are shown for $S_{\mathrm{crit}} = 1.2$ and $S_{\mathrm{crit}} = 1.4$, in both cases assuming $f_{\mathrm{act}} = 10\%$. Note that for the free running experiments, the confidence levels are calculated using a Welch's t-test instead of a paired-sample t-test as in the nudged ones.

Our results partly support the findings of Liu et al. (2009), who also reported a change of sign in the aviation-soot cirrus effect from negative to positive when increasing the critical saturation ratio from 1.2–1.3 to 1.4, although our RF values are considerably smaller, possibly due to the much lower active fraction assumed in this study: 0.1–10% compared to 100%. Systematically larger negative RFs are found also by the studies of Penner et al. (2009), Zhou and Penner (2014), Penner et al. (2018) and Zhu and Penner (2020), all assuming generally good ice nucleation abilities for aviation soot. Our study therefore
agrees with these in terms of the sign of aviation-soot effect, but there is a clear disagreement in the magnitude. This could be due to the details of the cloud microphysical parametrizations, to differences in the model setups (e.g., in the use of nudging) or in the representation of the vertical velocity, which can have a large impact on the resulting effects, as shown by our idealized sensitivity experiments. The model setup adopted here is very close to the one used in Hendricks et al. (2011), who used a similar base model (ECHAM4) and the same cirrus parametrization (Kärcher et al., 2006). They nevertheless found a non-
significant impact of aviation soot on natural cirrus clouds, which can possibly be ascribed to the use of a free-running model setup in that study, while the nudging mode used here likely helped to extract a statistically significant information from the model. Non-significant results were also found by Gettelman and Chen (2013) and, more recently, by McGraw et al. (2020). We note that for radiative forcings of the considered magnitude a statistically significant signal quantification is only possible with simulations using the nudging technique, although it is not completely clear how strong this procedure may impact on the
resulting effective radiative forcing values.



This work helps to disentangle and quantify the main uncertainties in the aviation-soot cirrus effect, but its actual magnitude (and to a lesser extent also its sign) remain uncertain. Our model simulations show that a more precise characterization of the ice nucleating properties of aviation soot could help to constrain at least the sign of the resulting RF. Further laboratory measurements are therefore needed, in particular concerning the role of cloud processing. In situ measurements are also essential,

in order to characterize the microphysical properties of the population of aviation soot, like number and size. On the modelling side, increasing the models intercomparability, e.g. by performing commonly-designed experiments in the context of an inter-comparison exercise, would provide valuable insights on the results and could help interpreting the model diversity in a better detail. In addition to the CMIP activities (Eyring et al., 2016; Collins et al., 2017), notable examples of the advantages of such exercises are provided by the AeroCom community (e.g., Schulz et al., 2006; Quaas et al., 2009; Mann et al., 2014; Samset

et al., 2014, see also https://aerocom.met.no/).

The EMAC-MADE3 model adopted here was shown to provide a satisfactory representation of aerosol, aerosol-induced ice formation in cirrus and overall cloud and radiation properties, but biases still exist. The model tends to overestimate the ice crystal number concentration in cirrus clouds at higher cirrus temperatures ($\gtrsim$225 K). Hence, the contribution of cirrus clouds formed in-situ by heterogeneous and/or homogeneous ice nucleation and cirrus clouds resulting from transport of ice from

510 the mixed phase (e.g., via detrainment from convective clouds) should be further explored. The representation of background aerosol like soot and mineral dust was already demonstrated to be quite good by Kaiser et al. (2019), Righi et al. (2020) and Beer et al. (2020). Nevertheless improvements are still possible, especially with the support of in situ measurements. The inclusion of additional background INP types, such as ammonium sulfate and glassy organics, in the model should also be considered, as this might change the outcome of the competition for available supersaturated water vapour among the different

INPs, with possible impacts on the aviation-soot effect estimated in this work.

*Code and data availability.* MESSy is continuously developed and applied by a consortium of institutions. The usage of MESSy and access to the source code is licensed to all affiliates of institutions which are members of the MESSy Consortium. Institutions can become members of the MESSy Consortium by signing the MESSy Memorandum of Understanding. More information can be found on the MESSy Consortium Website (http://www.messy-interface.org, last access: 3 March 2021). The model configuration discussed in this paper has been

developed based on version 2.54 and will be part of the next EMAC release (version 2.55). The exact code version used to produce the result of this paper is archived at the German Climate Computing Center (DKRZ) and can be made available to members of the MESSy community upon request. The output of the model simulations discussed in this work will be made available via doi in the final version of this paper.

*Author contributions.* MR conceived the study, designed and performed the simulations, analysed the data and wrote the paper. JH conceived the study and contributed to the model configuration, to the interpretation and to the text. CB contributed parts of the model code and

525 configuration, to the interpretation and to the text.



*Competing interests.* The authors declare no competing interests.

*Acknowledgements.* We are grateful to Marius Bickel and Patrick Jöckel (DLR, Germany), for contributing to the model improvements in the EMAC version used for this study. This manuscript has greatly benefited from helpful discussions with Ulrike Burkhardt, Klaus Gierens, Michael Ponater (DLR, Germany), Ulrike Lohmann and Colin Tully (ETH, Switzerland). The model simulations and data analysis for this
work used the resources of the Deutsches Klimarechenzentrum (DKRZ) granted by its Scientific Steering Committee (WLA) under project ID bd0080.

*Financial support.* This study was supported by the DLR aviation research program (Eco2Fly project), by the DLR transport research program (TraK project), by the DLR space research program (MABAK project) and by the European Commission via their Horizon 2020 Research and Innovation Program under Grant Number 875036 (ACACIA project).



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
