# Peer review of "Exploring the uncertainties in the aviation soot-cirrus effect"

_Atmospheric Chemistry and Physics, 2021_

## Referee Comment (RC2)

This study examined the sensitivity of microphysical property of cirrus clouds and radiation changes induced by aircraft soot. This manuscript was written very well and reviewed comprehensively. Lots of sensitivity experiments for ice nucleation ability, vertical velocity and nudging were conducted. However, I still have some suggestion to improve this study and make the conclusion clearer. The description of the model setup is not detail enough. For example, the method to determine the homogeneous freezing and its critical condition should be added. In addition, I suggest showing the ICNC from homogeneous nucleation, which is important to determine the RF of aircraft soot and its sensitivity. Else, what's the mechanism of the growth of aircraft soot? Is coating on aircraft soot considered? Will coating influence the ice nucleation ability of aircraft soot?

Figure 5 summarized the results of series of sensitivity experiments. However, the reason and mechanism leading to those results is not clearly explained, so that we still can not address exact conclusions and mechanisms from these experimental results. I suggest pay more attention to the mechanism explaining the sensitivities, especially those non-monotonous variations.

Although this study does not mean to provide an updated estimate on the aviation soot-cirrus effect. I would still suggest adding some comparison between observations (ICNC, radiation, IWC) and simulations, so that we can evaluate which sensitivity may be closer to actual conditions. Otherwise, we can not find any reference to determine the sensitivity parameters. If the simulated conditions are far away from real conditions, the sensitivity experiments would make no sense to address a general conclusion and help modeler to work further.

Specific comments:

1. Line 70: Zhu and Penner have published a corrigendum in Atmospheric Chemistry and Physics which changed the $S_{crit}$ to 1.35
2. Line 98: It is not clear that what are "such methods" did you mean here? Parameterization of vertical updraft?
3. Line 142: Actually, all models using the ice nucleation parameterization of Liu and Penner (2005) have temperature dependence, although not very exact.
4. The section 2 is like an introduction, which is some similar with the Section 1. I suggestion merging section 2 into Introduction secton.
5. Line 178: Why did you change the minimum CDNC to 50cm-3? Did you have any reference?
6. Line 183: Was the ERF here attributed by the changes only in warm clouds? Or combined with ERF on cirrus clouds?
7. Line 195: BCtag tracer refers to BC. So did you use the emission inventory of aircraft BC or aircraft soot? Do aircraft BC and soot have same ice nucleation ability and size distribution?

8. Table 1: The background soot is not very important for the ice nucleation and RF of aircraft soot since the number of INPs from background soot should be very small. Instead, I think dust could be mor important to influence heterogeneous nucleation since the number of INPs from dust are usually larger than soot. I would suggest adding some sensitivity experiment with different treatment to dust in addition to the sensitivity experiments of Scrit for background soot.

9. Line 265: As I saw, the connections between North America and East Asia are also marked.

10. Figure 3 c&e: I saw the mass concentration is high around 200hPa over the Antarctic, while that is low over the Southern tropics. However, the number concentration is opposite. In addition, the mass concentration is some high above 200hPa, but the number concentration is very low there. Could you please explain the mechanism of aircraft soot growing and the way to determine the number concentration? What are the size distributions of aircraft soot over tropics and polar area? Why are they largely different?

11. Line 287: Why the RF is positive when the critical saturation is high? Even though the nucleation efficiencies is low, INPs from aircraft soot also suppresses the homogeneous nucleation leading to small negative RF. Figure 5h indicates that the cloud frequency increases when Scrit is median and high, which was used to explain the enhanced cloud lifetime and positive RF. However, the cloud frequency increase only when fact=0.1% and the RF when fact=0.1% is much less positive than the cases with fact=1% and 10% (Scrit=1.4). In addition, the total ICNC increases but ICNC from homogeneous freezing decreases when fact=0.1, although the change in the ICNC from heterogenous nucleation did not show. Why do ICNC from homogeneous freezing only increase when fact=1% and Scrit=1.2&1.3(Figure 5e)?

12. Line 328: As you explained, the additional aircraft soot in the Southern Hemisphere could lead to enhanced heterogeneous nucleation thus positive RF. Why the ICNC did not increase when much more INPs added when Scrit=1.2? What's the different mechanism?

13. Line 345: Could you please show the changes in low-level clouds maybe in SI? So that we can know how much influence on the low-level clouds and contribution to the changes in shortwave.

14. Line 385: I don't understand why the RF of aviation soot is more negative and significant when the impact of aviation soot is reduced?

15. Line 420: Why does the longwave RF switch from cooling to warming?

---

## Author Comment (AC1)

**Reply to "Comment on acp-2021-329"**
**Mattia Righi, Johannes Hendricks and Christof G. Beer**

We are grateful to Bernd Kärcher for his comment and his suggestions on how to further improve the model-based estimates of the aviation soot-cirrus effect.

We are fully aware of the limitations related to the representation of the ice formation processes in our model (and in global models in general) and of the uncertainties connected with that. These are clearly mentioned and discussed in our study and are exactly the motivation for it. This is stated already in the title: *Exploring the uncertainties of the aviation soot-cirrus effect*. So, this paper is not simply *"the latest attempt to arrive at a GCM-based solution"*, as stated in Bernd Kärcher's comment, but it is a detailed analysis of which role the aforementioned limitations and uncertainties play in the quantification of the resulting radiative forcing. This is explicitly written in the introduction: "*Rather than attempting to provide a single estimate of the aviation soot-cirrus effect, the goal of this study is to explore the uncertainties related to the microphysical and dynamic aspects of this effect, in order to provide a realistic, albeit broad, range of possible values for the resulting climate impact.*" (lines 87-91). We also note that GCMs (or global models in general) are the only available tools to provide global quantifications of the radiative forcing resulting from this effect.

The comment seems to suggest that wrong parameters were chosen to drive the cirrus parametrization in this study (*"...I recommend using more appropriate estimations of both, activated fraction and nucleation threshold. The former happens to be greatly overestimated while the latter is underestimated in the present study."*). This is however not completely correct: as clearly shown in Fig. 1, we did not choose single values for these parameters, but we performed many model experiments to cover a wide range of measured values, for both the active fraction ($f_{act}$) and the nucleation threshold ($S_{crit}$), resulting in 9 different pairs for these two parameters. We also consciously included parameter combinations which might somewhat be less realistic, in order to directly compare with previous model studies where such values were adopted. We further note that the parameters deemed as "appropriate" by Bernd Kärcher in his comment and in his recent study (Kärcher et al., 2021) are fully covered by our analysis ($f_{act}$<1% and $S_{crit}$=1.4, i.e., close to the homogeneous freezing limit, see Fig. 1). Choosing larger values for $S_{crit}$ (e.g., $S_{crit}$=1.5) would very likely lead to results lacking any statistical significance. Hence, we refrained from using such values, also to avoid wasting large computing resources, which are required to perform such climate model simulations.

In spite of its known limitations, the cirrus parametrization adopted in our model (Kärcher et al., 2006) is operationally included in several other climate models and has been evaluated and applied in many peer-reviewed studies of the last 10 years (Hendricks et al., 2011; Kuebbeler et al. 2014; Gasparini and Lohmann, 2016; Gasparini et al., 2017; Penner et al., 2018; Righi et al., 2020; Gasparini et al., 2020; Lohmann et al., 2020). It therefore represents the state-of-the-art of global modelling of cirrus clouds. The applicability of this parametrization for the purpose of our study has been demonstrated in a very detailed model description and evaluation paper on the EMAC-MADE3 model (Righi et al., 2020).

Climate models, in many cases, need to deal with simplified representations of processes, in order to be applicable to the large spatial and temporal scales. This is a necessary compromise in order to describe aerosol-cloud-climate interactions on the global scale and to come up with a quantification of the resulting climate forcing (in this case, of the aviation-soot cirrus effect). We are grateful to Bernd Kärcher and his co-authors for their recent advancements in the microphysical description of the underlying effect (Kärcher et al., 2021; Marcolli et al., 2021) and we agree that they should be considered for implementation in future model studies. However, the implementation of new processes in a global climate model requires considerable efforts. This is especially the case when aerosol-cloud interaction processes are involved, since this requires a new tuning of the model, which is usually computationally expensive, and a re-evaluation of the whole system. Before further increasing the model complexity, it is therefore important to show whether the processes under consideration lead to relevant climate impacts. This is precisely the goal of our study, which can also serve as a basis of future, more refined studies, should some missing processes turn out to be relevant for the aviation effect under investigation.

As mentioned above, another goal of our paper is to directly compare with existing model-based quantifications of the aviation soot-cirrus climate effect, which in several cases reported very large radiative forcing values, as also criticized in the Kärcher et al. (2021) study. Part of the parameter space explored in our sensitivity simulations cover indeed the assumptions done in such studies, but nevertheless we were not able to confirm the large radiative forcing values reported there (see Sect. 5, second paragraph, lines 521-536). Hence, our conclusions are very much aligned to the conclusions of Kärcher et al. (2021), i.e., that in many of the existing model-based quantifications, the global radiative forcing effect via aviation-soot cirrus interactions may be overestimated.

In the revised version of our manuscript we have considered Bernd Kärcher's comments and the perspectives offered by the new studies he mentions. The following paragraphs have been added in different sections:

- **Sect. 3**: *"…which implies that only a very low fraction of aviation soot can effectively be active as INPs in the upper troposphere. This was also the conclusion by Kärcher et al. (2021): In a recent process-oriented analysis using a high-resolution column model based on the parametrization by Marcolli et al. (2021), they suggested that less than 1% of aviation soot particles can lead to the formation of ice crystals in competition with homogenous freezing."*
- **Sect. 4.2**: *"Kärcher et al. (2021) also argued against large RF effects from aviation soot-cirrus interactions, pointing to the limitations of global models in representing key processes as a possible reason for overestimated effects."*
- **Sect. 5 (Conclusions)**: *"Recent advancements in the representation of key microphysical processes for the investigated effects (Kärcher et al., 2021; Marcolli et al., 2021) should also be considered for implementation in global models. […] Moreover, the cirrus parametrization by Kärcher et al. (2006) implemented in EMAC-MADE3 has some known limitations which should be addressed in future versions of the model, for example replacing critical supersaturation with an activation spectrum following the change in the active INP population with increasing supersaturation and considering the effects of particle size and coating on the soot nucleation process."*

We believe that this kind of discussions greatly helps the scientific progress on a highly debated subject and we are grateful to ACP for the opportunity of publicly hosting these contributions.

**References**

Gasparini, B. and Lohmann, U.: Why cirrus cloud seeding cannot substantially cool the planet, *J. Geophys. Res. Atmos.*, 121, 4877-4893, https://doi.org/10.1002/2015jd024666, 2016.

Gasparini, B., Münch, S., Poncet, L., Feldmann, M., and Lohmann, U.: Is increasing ice crystal sedimentation velocity in geoengineering simulations a good proxy for cirrus cloud seeding?, *Atmos. Chem. Phys.*, 17, 4871-4885, https://doi.org/10.5194/acp-17-4871-2017, 2017.

Gasparini, B., McGraw, Z., Storelvmo, T., and Lohmann, U.: To what extent can cirrus cloud seeding counteract global warming?, *Environ. Res. Lett.*, 15, 054002, https://doi.org/10.1088/1748-9326/ab71a3, 2020.

Hendricks, J., Kärcher, B., and Lohmann, U.: Effects of ice nuclei on cirrus clouds in a global climate model, *J. Geophys. Res. Atmos.*, 116, 620, https://doi.org/10.1029/2010jd015302, 2011.

Kärcher, B., Hendricks, J., and Lohmann, U.: Physically based parameterization of cirrus cloud formation for use in global atmospheric models, *J. Geophys. Res. Atmos.*, 111, https://doi.org/10.1029/2005jd006219, 2006.

Kärcher, B.; Mahrt, F. & Marcolli, C.: Process-oriented analysis of aircraft soot-cirrus interactions constrains the climate impact of aviation, *Commun. Earth Environ.*, 1, 2, https://10.1038/s43247-021-00175-x, 2021.

Kuebbeler, M., Lohmann, U., Hendricks, J., and Kärcher, B.: Dust ice nuclei effects on cirrus clouds, *Atmos. Chem. Phys.*, 14, 3027-3046, https://doi.org/10.5194/acp-14-3027-2014, 2014.

Lohmann, U.; Friebel, F.; Kanji, Z. A.; Mahrt, F.; Mensah, A. A. & Neubauer, D. Future warming exacerbated by aged-soot effect on cloud formation. *Nat. Geosci.*, 13, 674-680, https://10.1038/s41561-020-0631-0, 2020.

Marcolli, C., Mahrt, F., and Kärcher, B.: Soot PCF: pore condensation and freezing framework for soot aggregates, *Atmos. Chem. Phys.*, 21, 7791-7843, https://doi.org/10.5194/acp-21-7791-2021, 2021.

Penner, J. E., Zhou, C., Garnier, A., and Mitchell, D. L.: Anthropogenic Aerosol Indirect Effects in Cirrus Clouds, *J. Geophys. Res. Atmos.*, 123, https://doi.org/10.1029/2018jd029204, 2018.

Righi, M., Hendricks, J., Lohmann, U., Beer, C. G., Hahn, V., Heinold, B., Heller, R., Krämer, M., Ponater, M., Rolf, C., Tegen, I., and Voigt, C.: Coupling aerosols to (cirrus) clouds in the global EMAC-MADE3 aerosol–climate model, *Geosci. Model Dev.*, 13, 1635-1661, https://doi.org/10.5194/gmd-13-1635-2020, 2020.

---

## Author Comment (AC2)

***Exploring the uncertainties in the aviation soot-cirrus effect***
**Mattia Righi, Johannes Hendricks and Christof G. Beer**
**Replies to referees'comments**

We are grateful to the referees for their useful suggestions and constructive criticisms which contributed to improve the manuscript. A detailed answer to each comment is provided below (referees' comments in *italic red*, authors' replies in black, quotes from the text in *italic blue*).

Please note that due to the insertion of two new figures in the revised version (Fig. 3 and Fig. 9), the numbering of all subsequent figures has changed with respect to the first version of the manuscript. In the replies below, we are always referring to the figure numbering of the revised version and specify both numberings in case of ambiguity.

**Reply to Reviewer #1 (Andrew Gettelman)**

*1. The manuscript could do a bit better at highlighting the importance of the model base state of clouds for the soot effect, and the balance of homogeneous v. heterogeneous ice nucleation. This seems to be a critical factor. Maybe the EMAC-MADE3 values could be compared to other previous work (e.g. similar to figures 4 and 5 in Gettelman et al 2012).*

Thanks for this suggestion, this is indeed an interesting aspect to look at. We extended Section 3 by adding a paragraph at the end to discuss the balance of homogeneously- and heterogeneously-formed ice crystals in the BASE simulation. An additional figure has been included (Fig. 3) to support the discussion, which also includes panels for IWC and ICNC in the BASE simulation, thus highlighting the model base state of clouds as suggested.

The new paragraph in Sect. 3 relates very well to the discussion on the model's representation of the vertical velocities (given in the previous paragraphs) and to Fig. 2: *"We further analyse the model mean state and, in particular, the impact of the vertical velocity on the ice formation processes in Fig. 3, depicting IWC and ICNC distributions in the BASE experiment, as well as the frequency of occurrence of homogeneous freezing events. The latter is estimated by considering the number concentration of homogeneously- and heterogeneously-formed ice crystals as diagnosed by the cirrus parametrization in the model. For each model grid-box and output time-step (11-h frequency, see R20), we then assign a value 1 (0) where ICNC from homogeneous freezing is larger (smaller) than ICNC from heterogeneous freezing. By averaging this over the whole simulation time-period (2001-2015), we obtain an estimate of the frequency of occurrence of homogeneous freezing events. IWC has its largest values (2–5 mg kg$^{-1}$) along the tropopause at all latitudes (Fig. 3a), while at 250 hPa the maxima are mostly in the Tropics (Fig. 3b). A similar pattern is followed by ICNC (Fig. 3a, b), with slightly higher concentrations (200–500 L$^{-1}$) in the Northern Extra-tropics, possibly driven by anthropogenic and dust emissions contributing large concentration of INPs at these latitudes. The zonal mean profile of the homogeneous freezing fraction (Fig. 3e) shows a large contribution (≥50%) of homogenous freezing at lower levels in the polar regions and close to the tropopause at all latitudes, with larger values in the extra-tropics than in the tropics. As expected, the geographical distribution at 250 hPa (Fig. 3b) closely matches the pattern of vertical velocities, in particular the strong updrafts of several tens centimeters per second induced by orographic waves (Fig. 2c) leads to enhanced homogeneous ice formation, with larger fractions found in the vicinity of the highest mountain ranges over the globe. Note that, due to the parametrization design, the analysis of the*

*homogeneous freezing fraction only counts pristine crystals, hence it only applies to the formation process (disregarding for instance the different residence times of ice crystals of different sizes)."*

*2. It would be nice to dive into the microphysics a bit. Could you perhaps show sensitivity plots for Figure 3 (especially zonal mean mass and number concentrations) and maybe ice crystal number and mass? This would be very helpful at understanding radiative forcing.*

As mentioned in the reply to the previous comment, we extended Sect. 3 and added the new Fig. 3 to discuss the balance between homogeneous and heterogeneous ice nucleation. Unfortunately, we do not have prognostic tracers for homogeneously- and heterogeneously-formed ICNC in the model. For the way the cloud scheme is designed, the number concentrations of homogeneously and heterogeneously formed pristine ice crystals are available as diagnostic quantities after each call to the Kärcher et al. parametrization. These diagnostic tracers experience only depositional growth in the model, while the other relevant processes (aggregation, accretion, and transport) are applied to the prognostic tracer for total IC concentration (see Kuebbeler et al., 2014; Righi et al., 2020). We therefore adopted the method outlined in the previous comment to estimate the homogeneous freezing fraction. The same analysis directly comparing the ICNC from the two formation processes would be biased by the fact that a single homogeneous freezing event can produce orders of magnitude more ice crystals than a heterogeneous freezing event.

Nevertheless, the homogeneous freezing fraction shown in the new Fig. 3 turned out to be an additional interesting metric for interpreting the aviation-soot cirrus effects. So, we decided to use this metric instead of the ICNC from homogenous freezing in Figs. 6, 8, and 9, and in the geographical maps in Fig. S7. This metric shows a generally higher statistical significance, thus facilitating the interpretation of the results. The text in Sect. 4.2 and 4.3 has been adjusted accordingly.

We do not think that additional sensitivity plots for Fig. 4 (previous Fig. 3) would provide additional information, since we did not vary the microphysical properties of aviation soot particles in our experiments, but just their ice nucleation abilities in the cirrus parametrization. Hence the distributions show in Fig. 4 apply to all experiments (except for a constant scaling factor $f_{act}$ for aviation soot). The resulting impacts on ICNC and total water are already shown in Fig. 6 (previous Fig. 5). The low statistical significance of the aviation impact on ICNC unfortunately prevents any robust analysis on their geographical distribution (which is therefore not shown).

*For vertical velocity perturbations this is all buried in Figure S7, but would probably be more useful to try to put in the main text for this as well as for the different Script and f values. Maybe the relevant properties from figure S7 for each case could be reduced to a zonal mean perturbation, and then the line plot would enable all sensitivity studies to be put on one plot. Clearsky and total water are probably not that relevant. That would leave a 6 panel figure with all the relevant information. One for W perturbations and one for the Scrit and F perturbations.*

Thank you for this valuable suggestion. This is indeed an optimal way to summarize the vertical velocity experiments in a single figure. This is now Fig. 9, which complements Fig. 8 for the same variables. The discussion in the text has been extended accordingly.

We also tried a similar solution for the $S_{crit}/f_{act}$ sensitivities, but this resulted in a very crowded plot (9 lines). Since in this case we are exploring a two-dimensional parameter space, we believe that presenting the results in a matrix structure is more intuitive and facilitates the comparison between Figs. 5-7 and S3-S7, also in connection with Fig. 1. Furthermore, the statistical significance is shown more explicitly, which is crucial for the interpretation of the results.

*3. In addition, can you state what the contrail RF is for just H2O without aerosol perturbations? This would be a useful baseline to compare to other work (e.g., estimates in Lee et al 2021). Maybe this is in Righi et al 2020, but please state it here and note how it is similar (or not) to other work.*

The role of contrails is an important issue but it cannot be explored with the current model configuration. Work is ongoing as part of a PhD project to extend the Kuebbeler et al. (2014) cloud scheme of EMAC-MADE3 with a parametrization for contrail cirrus. The first simulations are being performed and a corresponding manuscript will be submitted soon. We aim at combining this new approach with simulations of aerosol-induced natural cirrus perturbations in the future.

*Page 1, L24: 'milden' is not an English word. Maybe 'reduce'. Also note the pandemic did temporarily reduce the increase. See Gettelman et al 2021.*

Word corrected and reference added, thank you.

*Page 2, L59:  no statistically significant effects were reported by Gettelman…*

Corrected.

*Page 4, L119: I don't think you mentioned this important point (or at least not sufficiently) in the introduction: that the model cloud state (humidity and water content and homo or heterogeneous nucleation dominance) may determine the response to soot as well. This has been discussed in the literature before, and should be highlighted in the introduction. See general comment above.*

The model mean state has been extensively documented and evaluated in Righi et al. (2020). All relevant information, including the model's sensitivity to tuning parameters, can be found in that paper. However, as suggested in one of the comments above, we added further plots in Fig. 3 to show IWC, ICNC and the balance between homogeneous and heterogenous freezing, also relating this to the representation of vertical velocity in Fig. 2.

*Page 7, L178: what do these gamma values mean? Please state what autoconversion scheme is being used. Is the minimum CDNC for the autoconversion scheme or independent?*

Good point. This has been discussed in Righi et al. (2020), but we added short explanations in the text for more clarity: *"…the enhancement factor $\gamma_r$ for the rate of rain formation by autoconversion has been reduced from $\gamma r = 8$ to $\gamma r = 4$, and the minimum CDNC (a threshold introduced in the cloud scheme to avoid unrealistically low concentrations in pristine regions) has been increased from 20 cm−3 to 50 cm−3."*. As requested by Reviewer #2, we also better motivated our choice of the minimum CDNC value (see below).

We use the autoconversion scheme by Khairoutdinov and Kogan (2000). Although we did not test other schemes, it is reasonable to assume that the tuning parameters will depend on the choice of the scheme as well, as the model was found to be quite sensitive to this process representation (see Righi et al., 2020, for details).

*Page 7, L195: aviation soot = BCtag?*

Yes, we stated this at the beginning of the paragraph: *"Another relevant improvement to the model configuration applied here is the introduction of an additional tracer BCtag to which the soot emissions from the aviation sector are assigned."*.

*Page 9, L230: is there a timescale for the nudging or is it replacement of the model dynamical fields?*

There is indeed a timescale for nudging: 6 h for wind vorticity, 24 h for temperature and surface pressure, and 48 h for wind divergence. We added this to the text. Thanks for noting this.

*Page 14, L289: What is the overall contrail RF with and without soot (I.e. just aviation H2O)? That helps put this study in context of other work. See general comment.*

As mentioned above, unfortunately it is not yet possible to estimate this with the current model configuration, but work is ongoing to extend the model in that sense.

*Page 14, L315: I'm not quite following this logic. How does enhanced cloud lifetime mesh with reduced cloud fraction?*

This sentence is indeed inconsistent, thanks for pointing this out. It has been rephrased as follows: *"For lower efficiencies (higher critical saturation ratios), the warming (reduced cooling) in the shortwave appears to be more important (possibly due to a reduction in cloud frequency, Fig. 6h) and leads to a slightly positive overall RF effect."*.

*Page 16, L352: how does this compare to the total contrail RF?*

We added a sentence in the conclusion (first paragraph) to put these numbers into the broader context of the aviation effects: *"For comparison, the estimated effective RF of the main climate forcers from aviation reported in the recent assessment by Lee et al. (2021) are 34.3 mW m$^{-2}$ (CO$_2$), 57.3 mW m$^{-2}$ (contrail cirrus), and 17.5 mW m$^{-2}$ (NO$_x$), with the total aviation effect amounting to 110.9 mW m$^{-2}$. Therefore, the aviation soot-cirrus effect quantified here potentially represents an important contribution to the aviation climate impact."*

*Page 17, L363: order of tens of mWm$^{-2}$*

Corrected.

*Page 18, L377: seems like the activation efficiency is key for disparity across studies. Can you intuit anything about model dynamics and cloud environment?*

In our view, the activation efficiency can only explain part of the disparity. Our variation study basically covers the whole space of assumptions made by previous studies, but still our estimates are significantly lower than in most of them (with the exception of Hendricks et al., 2011, and Gettelman and Chen, 2013). Uncertainties in the model dynamics could add another factor of about 1.7 to the quantified RF, but the disparities across models are of the order of a factor of 10. Hence, the models' base state (including cloud environment) is probably the key to understand the differences across models.

*Page 19, L407: pair of assumptions for the ice nucleation….*

Corrected, thank you.

*Page 20, L416: so the effect jumps around with sub grid W? That seems a bit worrisome. You explanation is plausible however that the effect changes sign of sensitivity with W.*

The fact that the dynamics plays such an important role is indeed worrisome, but not completely unexpected. However, it is good to see that our idealized studies show that it adds *only* about a factor 1.7 to the estimated RF and does not lead to a change of sign. The latter seems to be

controlled only by the microphysics (although we did not perform the vertical velocity studies with all pairs of $S_{crit}/f_{act}$).

*Page 23, L510: maybe you could plot homogenous IN fraction? This may be a good model metric for comparison? Can you comment any more here about the impact of the model base state on response to aircraft soot? Seems like you can draw some significant conclusions, basing the homo v. Hetero balance on your variation of w and $S_{crit}$?*

Thanks for this suggestion. As discussed above, we now show the homogeneous freezing fraction in the BASE simulation (Fig. 3) and discuss the aviation-induced changes in this metric for the sensitivity to $S_{crit}/f_{act}$ (Fig. 6) and to the vertical velocity (Figs. 8, 9 and S7). The text in the respective sections has been extended to discuss the impact of aviation on this metric.

**Reply to Reviewer #2 (anonymous)**

*The description of the model setup is not detail enough. For example, the method to determine the homogeneous freezing and its critical condition should be added.*

The model configuration has been very extensively documented in Righi et al. (2020). Here, we mainly focused on the new features specifically introduced to target the aviation effect and described in Sect. 3.

To address the specific question about homogeneous freezing, we added a paragraph explaining the main features of the cirrus parametrization used in the model: *"…following Kärcher et al. (2006). This parametrization considers the competition among the different ice formation processes in increasing order of critical saturation ratio, i.e. from the most efficient heterogeneous freezing process to homogeneous freezing. The increase of supersaturation driven by vertical updrafts proceeds until it is compensated by the loss of water vapour due to the growth of ice crystals (either freshly formed or pre-existing from the previous model time-step). For large enough supersaturations, both heterogeneous and homogeneous freezing can take place and their competition is accounted for in the model."*.

*In addition, I suggest showing the ICNC from homogeneous nucleation, which is important to determine the RF of aircraft soot and its sensitivity.*

Following a similar suggestion by reviewer #1, we added Fig. 3. This shows IWC and ICNC in the BASE experiment, together with the homogenous freezing fraction. We also extended the discussion at the end of Sect. 3 and related this to the vertical velocity analysis in the same section (see reply to the first comment of Reviewer #1). Furthermore, we now use the homogenous freezing fraction as an additional metric to analyse the aviation-soot effects in Figs. 7-9 and S7. As discussed above, this metric provides much more statistically significant results than the ICNC from homogenous nucleation, hence it makes the interpretation of the results more robust.

*Else, what's the mechanism of the growth of aircraft soot? Is coating on aircraft soot considered? Will coating influence the ice nucleation ability of aircraft soot?*

Yes, coating on aircraft soot is considered: as described in Kaiser et al. (2019) and Righi et al. (2020), the aerosol submodel MADE3 considers both insoluble (soluble coating <10% of modal dry mass) and mixed (coating >10%) soot and aircraft soot. This is also mentioned in Sect. 3: *"The BCtag tracer is distributed into the same 6 modes as the standard BC tracer of MADE3, namely Aitken, accumulation and coarse mode, each with insoluble and mixed states."*.

Whether coating can influence the ice nucleation ability of soot is an interesting question, which however cannot be answered by a global model. Laboratory studies show indeed that coatings could have an impact (e.g., Kanji et al. 2017, https://doi.org/10.1175/AMSMONOGRAPHS-D-16-0006.1; and Zhang et al., 2020, https://doi.org/10.5194/acp-20-13957-2020).

Due to the existing uncertainties, we did not attempt to include the effect of coating on the nucleation ability explicitly. We followed the alternative approach to vary the nucleation properties systematically. The different results discussed in the paper, however, should also cover cases where nucleation is suppressed by coating (e.g., low $f_{act}$, high $S_{crit}$).

*Figure 5 summarized the results of series of sensitivity experiments. However, the reason and mechanism leading to those results is not clearly explained, so that we still can not address exact*

*conclusions and mechanisms from these experimental results. I suggest pay more attention to the mechanism explaining the sensitivities, especially those non-monotonous variations.*

We disagree with the reviewer on this point. The whole Sect. 4.2 is dedicated to the discussion of these sensitivity experiments, and further support to the interpretation of the results is given in Fig. 7 (previous Fig. 6) and in the maps in the supplement (Figs. S2-S6). It is also not clear to us why monotonous variation should be expected from an effect which results from the interplay of many different microphysical and dynamic processes, most of which are highly non-linear.

*Although this study does not mean to provide an updated estimate on the aviation soot-cirrus effect. I would still suggest adding some comparison between observations (ICNC, radiation, IWC) and simulations, so that we can evaluate which sensitivity may be closer to actual conditions. Otherwise, we can not find any reference to determine the sensitivity parameters. If the simulated conditions are far away from real conditions, the sensitivity experiments would make no sense to address a general conclusion and help modeler to work further.*

The sensitivity of the overall ICNC and IWC to the ice nucleating abilities of aviation soot are far too small and well within the observational uncertainties to draw any kind of robust conclusion from such a comparison (as it is also clear from Fig. 6). That is one of the main challenges in this kind of studies.

As we argued in the conclusions, we believe that further insights into this effect could be gained from laboratory measurements specifically focusing on the ice nucleating abilities of aviation soot. Our study aims at providing a plausible range for the aviation soot-cirrus RF. As long as more data from laboratory measurements become available, it should be possible to further constrain the parameter space shown in Fig. 4 (previous Fig. 3) and provide more precise estimates. Aircraft measurements of cirrus properties (e.g., Krämer et al., 2020, used in Righi et al. 2020) are extremely helpful to evaluate model base state and improve its overall performance, but they are not detailed enough to constrain the impact of aviation soot only. In our view, this is something only global models can do, at the moment.

*1. Line 70: Zhu and Penner have published a corrigendum in Atmospheric Chemistry and Physics which changed the Scrit to 1.35*

Thank you for pointing us to this corrigendum! This is very relevant for Fig. 1 and for the introduction, which have been corrected accordingly.

*2. Line 98: It is not clear that what are "such methods" did you mean here? Parameterization of vertical updraft?*

We refer to the parametric approach. We have rephrased the sentence to make this clearer: *"On a more general level, such parametric approaches were successfully used…"*.

*3. Line 142: Actually, all models using the ice nucleation parameterization of Liu and Penner (2005) have temperature dependence, although not very exact.*

Thank you for pointing this out. We have corrected the sentence in this way: *"Furthermore, the experimental results show a clear temperature dependence of the soot ice nucleating properties, which most of the model parametrizations do not take into account (an exception here is the Liu and Penner (2005) parametrization)."*.

*The section 2 is like an introduction, which is some similar with the Section 1. I suggestion merging section 2 into Introduction.*

Section 2 actually focuses on a specific issue (the uncertainties in the nucleation abilities of aviation soot) and is therefore not suitable for the introduction (which usually covers the overall topic in general). Furthermore, Section 2 introduces and discusses the first figure of the paper, which would be a quite uncommon practice for an introduction section. So, we would prefer to keep these two sections separated, also in the interest of overall readability.

*Section 5. Line 178: Why did you change the minimum CDNC to 50 cm$^{-3}$? Did you have any reference?*

As explained in the same paragraph, a minor improvement in the time integration of the cloud prognostic variables required a retuning of the model, and the minimum CDNC is one of the tuning parameters adopted here (see Righi et al., 2020, for details). We added three references in the text to support our choice for this value: *"This value is close to the 40 cm$^{-3}$ value used by Neubauer et al. (2019) in the ECHAM6 model and is supported by typical CDNC values measured by satellite in pristine regions (Bennartz and Rausch, 2017; Grosvenor and Wood, 2018)."*.

*Line 183: Was the ERF here attributed by the changes only in warm clouds? Or combined with ERF on cirrus clouds?*

The ERF value reported here refers to the total anthropogenic aerosol effect. We rephrased the sentence for more clarity.

*Line 195: BCtag tracer refers to BC. So did you use the emission inventory of aircraft BC or aircraft soot? Do aircraft BC and soot have same ice nucleation ability and size distribution?*

The use of the term *soot* in aviation research is often ambiguous, as we explicitly noted at the end of that paragraph: *"To avoid confusion, we note here that the MADE3 BC and BCtag tracers actually refer to black carbon, i.e. an aerosol type composed only of carbon, but we are using the term soot in this paper for consistency with most of the literature on aviation effects, although these definitions are not fully consistent (see Petzold et al., 2013, for a more detailed discussion on this terminology)."*. The CMIP6 emission inventory used in this study reports BC emissions for all sectors (including aviation), see Gidden et al. (2018).

*Table 1: The background soot is not very important for the ice nucleation and RF of aircraft soot since the number of INPs from background soot should be very small. Instead, I think dust could be more important to influence heterogeneous nucleation since the number of INPs from dust are usually larger than soot. I would suggest adding some sensitivity experiment with different treatment to dust in addition to the sensitivity experiments of S$_{crit}$ for background soot.*

This is correct. Not only dust, but also other INP types (like crystalline ammonium sulfate) could play a significant role in the competition for ice formation. We are currently investigating additional INP types as part of a follow-up study, which could include further sensitivity experiments on the role of dust, as suggested.

*Line 265: As I saw, the connections between North America and East Asia are also marked.*

Yes, they are, but less than over the North Atlantic (between Europe and North America), as we stated in that sentence.

*Figure 3 c&e: I saw the mass concentration is high around 200hPa over the Antarctic, while that is low over the Southern tropics. However, the number concentration is opposite. In addition, the mass concentration is some high above 200hPa, but the number concentration is very low there. Could you please explain the mechanism of aircraft soot growing and the way to determine the number concentration? What are the size distributions of aircraft soot over tropics and polar area? Why are they largely different?*

The mechanisms of aircraft soot growing are the same as for background soot as stated in Sect. 3: *"Another relevant improvement to the model configuration applied here is the introduction of an additional tracer BCtag to which the soot emissions from the aviation sector are assigned. The BCtag tracer is distributed into the same 6 modes as the standard BC tracer of MADE3, namely Aitken, accumulation and coarse mode, each with insoluble and mixed states. The BC and BCtag tracers have the same physical properties and undergo exactly the same processes in the model…"*. See Kaiser et al. (2014, 2019) for a detailed explanation of the MADE3 microphysics, including particle growth.

The way to determine number concentration is explained in detail in Sect. 3: *"To calculate the number concentration of INPs for the different types we use the same approach as R20, while for the newly introduced BCtag tracer we derive the number concentration from the tracer mass, by assuming aviation soot to follow the bimodal size distribution measured by Petzold et al. (1999) in the plume of a B737-300 aircraft. This distribution is characterized by median diameters of 25 and 150 nm, and geometric standard deviations of 1.55 and 1.65, for the Aitken and accumulation modes, respectively."*.

The reasons for the difference between mass and number distributions is discussed in Sect. 4.1: *"The reason for such sharply structured patterns is that particles in the Aitken mode, which dominate total particle number, are characterized by a shorter lifetime due to particle-particle interactions, which effectively reduce their number concentration away from sources, while their mass is of course conserved."*.

*Line 287: Why the RF is positive when the critical saturation is high? Even though the nucleation efficiencies is low, INPs from aircraft soot also suppresses the homogeneous nucleation leading to small negative RF.*

The suppression of homogeneous nucleation leads in general to a positive RF due to the inverse Twomey effect. However, as explained in the following paragraph, this is more complicated and the RF values shown in Fig. 5 are the result of the combination of shortwave and longwave forcing, as shown in Fig. 6a-d. The shortwave RF is positive for all investigated cases, while the longwave one is negative, with a very strong contribution from the clear-sky part for $S_{crit}$=1.2. This contribution strongly decreases for $S_{crit}$>1.2, not counteracting the shortwave warming anymore and resulting in an overall positive effect.

*Figure 5h indicates that the cloud frequency increases when $S_{crit}$ is median and high, which was used to explain the enhanced cloud lifetime and positive RF. However, the cloud frequency increase only when $f_{act}$=0.1% and the RF when $f_{act}$=0.1% is much less positive than the cases with $f_{act}$=1% and 10% ($S_{crit}$=1.4).*

Actually, Fig. 5h (now 6h) shows a decrease in cloud frequency, which explains the positive shortwave RF. The statistical significance of the results, however, poses a challenge to the interpretation: statistically significant changes in the cloud frequency in Fig. 6h are found only for $f_{act}$=10%, and for lower $f_{act}$ values only at $S_{crit}$=1.2. Considering only the statistically significant values, this is consistent with the changes in the shortwave RF (Fig. 6a), while no conclusions should be drawn from the non-significant cases.

Further note that these panels consider global mean values: since these include very different regimes of cirrus formation characteristics, the effects mostly reflect a superposition by different mechanisms. So, it is hard to explain global effects following only a specific chain of mechanisms. This is the reason why we complemented the analysis with the regional means in Fig. 7 and the maps plots in Figs. S2-S6.

*In addition, the total ICNC increases but ICNC from homogeneous freezing decreases when $f_{act}$=0.1, although the change in the ICNC from heterogenous nucleation did not show. Why do ICNC from homogeneous freezing only increase when $f_{act}$=1% and $S_{crit}$=1.2&1.3 (Figure 5e)?*

As for the previous comment, we believe it is not possible to argue with changes in ICNC in Fig. 5 (now Fig. 6), given the very low statistical significance of the results. This is the reason why we focused on other cloud variables to interpret the RF effects we see in Fig. 5 and Fig. 6a-d. However, as discussed above, we replaced Fig. 6e and now show the aviation-induced changes in the homogeneous freezing fraction, which have a larger statistical significance and make the discussion in Sect. 4.2 more robust.

The model was configured to maximize the chances of obtaining statistically significant signals from aviation soot, by tagging aviation soot and using a nudging technique over a quite long simulation period (15 years). This, however, does not always guarantees statistically significant results for all variables. This is an intrinsic limitation of climate models that has to be accepted.

*Line 328: As you explained, the additional aircraft soot in the Southern Hemisphere could lead to enhanced heterogeneous nucleation thus positive RF. Why the ICNC did not increase when much more INPs added when $S_{crit}$=1.2? What's the different mechanism?*

A possible explanation is that homogeneous freezing could also be reduced in such situation, thus counteracting the INP-driven increase of ICNC. As mentioned in the reply to the previous comment, however, ICNC hardly show a statistically significant signal, which prevented us from interpreting the results along these lines.

*Line 345: Could you please show the changes in low-level clouds maybe in SI? So that we can know how much influence on the low-level clouds and contribution to the changes in shortwave.*

The model is configured in a way that aviation soot only impacts cirrus clouds, while no differences in the aerosol interactions with low-level clouds are considered in our experiments. To avoid misinterpretation of the results, we made this clearer by adding the following statement in Sect. 3: *"The difference between these two experiments hence provides an estimate of the climate impact of aviation soot on natural cirrus clouds, while excluding the effects of the interactions with clouds at lower levels (e.g., sulfate impact on liquid clouds)."*

*Line 385: I don't understand why the RF of aviation soot is more negative and significant when the impact of aviation soot is reduced?*

We could not understand this question, as there is no such statement at line 385. What we show in this study is rather the opposite: the RF of aviation soot is more negative only when very good ice nucleation abilities for soot are assumed ($S_{crit}$=1.2), with decreasing importance (in absolute term) for decreasing values of $f_{act}$ (see Fig. 5).

The statement at line 385 refers to a sensitivity experiment where we reduced $S_{crit}$ for *background soot* from 1.4 to 1.2. In this case, we see that the effect of *aviation soot* is reduced from −35.3 to −25.7 mW m$^{-2}$ (with similar statistical significance). The reason for this is mentioned in the same

paragraph: *"This was to be expected, since increasing the ice nucleation abilities of background soot enhances the competition with aviation soot and the other INPs for available water vapour, thus reducing the impact of aviation soot on natural cirrus clouds."*.

*Line 420: Why does the longwave RF switch from cooling to warming?*

Good point, we missed that in the discussion about Fig. 8. We extended the text as follows (also including references to the new Fig. 9): *"At larger vertical velocities, homogeneous freezing becomes very effective and rapidly consumes the available supersaturated water vapour, so that heterogeneously formed ice crystals have less time to grow to larger sizes and the sedimentation process becomes less effective. As shown in Figs. 8f and 9f, the aviation-induced reduction of the homogeneous freezing fraction is significantly smaller at $v \geq 20$ cm s$^{-1}$. As a consequence, in this somewhat extreme regime the impact of aviation soot on the main cloud variables (Figs. 8f-i and Figs. 9f-i) loses importance and results in smaller aviation-induced changes in the radiative fluxes (Figs. 8b-e and Figs. 9b-e), although their combined effect remains large (Fig. 8a). For the same reason, also the dehydration effect, which we found to induce a pretty strong clear-sky cooling in the longwave (Sect. 4.2), becomes less effective at higher vertical velocities and explains the change of sign at $v = 20$ cm s$^{-1}$ in Fig. 8e."*